# Intracranial human recordings reveal association between neural activity and perceived intensity for the pain of others in the insula

Efe Soyman[1,2†], Rune Bruls[1†], Kalliopi Ioumpa[1†], Laura Müller-Pinzler[3], Selene Gallo[1‡], Chaoyi Qin[1], Elisabeth CW van Straaten[4], Matthew W Self[5], Judith C Peters[5,6,7], Jessy K Possel[5], Yoshiyuki Onuki[8], Johannes C Baayen[9], Sander Idema[9], Christian Keysers[1,10*], Valeria Gazzola[1,10*]

[1]Social Brain Lab, Netherlands Institute for Neuroscience, Royal Netherlands Academy of Art and Sciences, Amsterdam, Netherlands; [2]Social Cognitive and Affective Neuroscience Lab, Koc University, Istanbul, Turkey; [3]Social Neuroscience Lab, Department of Psychiatry and Psychotherapy, University of Lübeck, Lübeck, Germany; [4]Department of Neurology and Clinical Neurophysiology, Amsterdam UMC, Vrije Universiteit Amsterdam, Amsterdam, Netherlands; [5]Department of Vision and Cognition, Netherlands Institute of Neuroscience, Royal Netherlands Academy of Art and Sciences, Amsterdam, Netherlands; [6]Department of Cognitive Neuroscience, Faculty of Psychology and Neuroscience, Maastricht University, Maastricht, Netherlands; [7]Maastricht Brain Imaging Center (M-BIC), Maastricht University, Maastricht, Netherlands; [8]Department of Neurosurgery, Jichi Medical University, Shimotsuke, Japan; [9]Department of Neurosurgery, VUmc, Amsterdam University Medical Center, Amsterdam, Netherlands; [10]Brain and Cognition, Department of Psychology, University of Amsterdam, Amsterdam, Netherlands

*For correspondence:
c.keysers@nin.knaw.nl (CK);
v.gazzola@nin.knaw.nl (VG)

†These authors contributed equally to this work

Present address: ‡Department of Psychiatry, Amsterdam University Medical Center, Amsterdam, Netherlands

**Abstract** Based on neuroimaging data, the insula is considered important for people to empathize with the pain of others. Here, we present intracranial electroencephalographic (iEEG) recordings and single-cell recordings from the human insula while seven epilepsy patients rated the intensity of a woman's painful experiences seen in short movie clips. Pain had to be deduced from seeing facial expressions or a hand being slapped by a belt. We found activity in the broadband 20–190 Hz range correlated with the trial-by-trial perceived intensity in the insula for both types of stimuli. Within the insula, some locations had activity correlating with perceived intensity for our facial expressions but not for our hand stimuli, others only for our hand but not our face stimuli, and others for both. The timing of responses to the sight of the hand being hit is best explained by kinematic information; that for our facial expressions, by shape information. Comparing the broadband activity in the iEEG signal with spiking activity from a small number of neurons and an fMRI experiment with similar stimuli revealed a consistent spatial organization, with stronger associations with intensity more anteriorly, while viewing the hand being slapped.

## Editor's evaluation

This fundamental work shows that insular broadband activity (20-190 Hz) correlates with the perceived intensity of others' painful experiences viewed as movies. This finding is not only important to the pain field but also constitutes an important contribution to the neuroscience

of empathy. However, whereas the intracranial recording data are compelling, a limitation of the study remains the lack of specific control stimuli to determine whether the early broad-band insular response when viewing the hand could be related to the observation of body movement rather than "intensity coding for the pain of others", and whether the late broadband response to a facial expression of pain might also be observed for other facial expressions.

## Introduction

Sharing the distress of others is central to empathy. fMRI studies show that a number of brain regions involved in the direct experience of pain also increase their activity while participants perceive the pain of others, including the cingulate cortex, the insula, and the somatosensory cortices (*Jauniaux et al., 2019*; *Keysers et al., 2010*; *Lamm et al., 2011*; *Timmers et al., 2018*). Across humans, primates, and rodents, lesions in these regions impair the perception and the sharing of others' emotions (*Paradiso et al., 2021*), providing evidence for their causal contribution to the perception or sharing of the emotions of others. A number of recent studies have used multivoxel pattern analysis to explore how these regions encode the pain of others using fMRI signals, with particular attention to the insula. *Krishnan et al., 2016* showed participants' images of hands or feet in painful or innocuous situations and found a pattern across voxels in the insula that could predict how much pain people reported they would feel in the depicted situations. *Corradi-Dell'Acqua et al., 2016* also reported that the pattern of insula activity could discriminate between trials in which a cue signaled that someone else was receiving a shock from nonshock trials. Finally, *Zhou et al., 2020* reanalyzed a dataset in which participants viewed photographs of hands in painful or nonpainful situations, or of painful and neutral facial expressions. They found that, in the insula, similar but dissociable patterns supported painfulness decoding for hands and faces: similar in that a pattern trained to discriminate painfulness from faces could do so from hands and vice versa, with the rostral insula contributing to both patterns; but dissociable in that many voxels contributed only to decoding of either faces or hands.

Directly recording electrical signals from these regions in humans would complement these more indirect fMRI measurements and sharpen our understanding of how these regions represent the intensity of other people's pain for at least two reasons. First, fMRI records a mixed signal that includes synaptic input and local neural processing. Localizing blood-oxygen-level-dependent (BOLD) activity that encodes a particular stimulus property thus cannot ensure that neurons in that region actually have spiking activity that encodes that property (*Boynton, 2011*). For instance, BOLD signals in V1 fluctuate based on whether a stimulus is perceived or not in binocular rivalry (*Boynton, 2011*; *Maier et al., 2008*). In contrast, simultaneous electrical recordings in V1 show that broadband gamma activity, which is tightly coupled to spiking (*Bartoli et al., 2019*; *Buzsáki et al., 2012*; *Miller et al., 2014*), responds to a stimulus equally well whether it is perceived or suppressed. Only the slower components, <20 Hz, that are known to carry feedback synaptic input fluctuate with perception (*Maier et al., 2008*). Being able to record electrical activity, particularly in the broadband gamma range, would thus be critical to localize where in this circuitry neuronal spiking indeed represents the pain of others. Second, fMRI's low temporal resolution makes it difficult to characterize the time course of responses.

For the anterior cingulate, we have intracranial recordings: *Hutchison et al., 1999* documented a single neuron in epileptic patients that responded to the sight of a finger being pin-pricked with increased firing rate, and a recent rodent study revealed that cingulate neurons responding to pain experience have responses that increase with the intensity of the pain experienced by another rat (*Carrillo et al., 2019*). In contrast, although the insula is central in the neuroimaging literature on empathy, and shows increases of BOLD signal for watching painful compared to nonpainful social stimuli (*Jabbi et al., 2007*; *Jauniaux et al., 2019*; *Lamm et al., 2011*; *Meffert et al., 2013*; *Singer et al., 2004*; *Timmers et al., 2018*; *Wicker et al., 2003*), and shows patterns that encode painfulness (*Corradi-Dell'Acqua et al., 2016*; *Krishnan et al., 2016*; *Zhou et al., 2020*), we still lack insular intracranial recordings while individuals witness the pain of others. Intracranial electroencephalography (iEEG) has been recorded in the insula during the self-experience of pain (*Bastuji et al., 2018*; *Bastuji et al., 2016*; *Liberati et al., 2020*), and the insula and adjacent SII are the only cortical regions where iEEG electrode stimulation can induce painful sensations (*Jobst et al., 2019*; *Mazzola et al., 2012*), but to our knowledge there are no published studies recording from insular electrodes while patients witness the pain of others. The degree to which neuronal activity local to the insula, as

opposed to feedback synaptic input from other regions such as the cingulate, encodes the intensity of other people's pain therefore remains unclear and the time course of such neural activity remains undercharacterized.

To fill this gap and characterize the electrophysiological responses of the insula to the pain of others, we collected depth electrode recordings from seven epileptic patients during presurgical exploration, while they rated the different intensities of pain they perceived in another person in a video (*Figure 1a and b*). All these patients had macroelectrodes in their insulae that yielded local field potentials (LFPs) capable of measuring broadband gamma activity (circles in *Figure 1c*). Three patients, additionally, had microelectrodes at the tip of some macroelectrodes to record from isolated insular neurons (pluses in *Figure 1c*). Our stimuli also included two ways in which pain is perceived in others (*Figure 1a*). Half the stimuli (Faces) showed a female receiving electroshocks on the hand and expressing pain through facial expressions (furrowing eyebrows and tightening eyes). The other half (Hands) showed the protagonist's hand slapped by a leather belt, and pain intensity had to be deduced from the movements of the belt and the hand. In both cases, movies, rather than static images, were chosen to provide richer and more ecological stimuli and provide information about the temporal dynamics with which such movies are represented in a field still dominated by the presentation of static images (*Adolphs et al., 2003*; *Zinchenko et al., 2018*). We used these two classes of stimuli because both tap into the visual perception of other people's pain, and we start to understand that they do so through partially overlapping and partially dissociable routes (*Jauniaux et al., 2019*; *Keysers et al., 2010*; *Timmers et al., 2018*). For instance, the Hand stimuli depend on the hand region of the somatosensory cortex (*Gallo et al., 2018*), while facial expressions depend on both the ventral somatosensory cortex and the insula (*Adolphs et al., 2000*; *Dal Monte et al., 2013*; *Mattavelli et al., 2019*), whereby the insula appears to show partially overlapping and partially discriminable patterns encoding painfulness for these two types of stimuli (*Zhou et al., 2020*).

Our aim is to localize insular recording sites that do or do not encode the perceived intensity of others' experiences from Face and/or Hand stimuli and to exploit the temporal resolution of iEEG to explore the timing of intensity coding and how it relates to the timing of intensity-relevant stimulus features. To provide evidence that a specific recording site does not encode intensity for a stimulus type, we supplement frequentist statistics with Bayesian statistics, which generate a Bayes factor (BF$_{10}$; *Keysers et al., 2020*). The Bayes factor indicates the probability of the data if there is intensity encoding (H$_1$) divided by that if there isn't (H$_0$). A BF$_{10}$ < ⅓, which indicates the data is at least three times more likely under H$_0$ than H$_1$, is considered evidence for the absence of intensity coding. Importantly, in Bayesian statistics, whenever possible, it is advised to commit to a directional hypothesis (H+) that endows the BF$_{+0}$ with higher sensitivity to detect evidence for both H+ and H$_0$ (*Keysers et al., 2020*; *Wagenmakers et al., 2016*). Given that fMRI experiments overwhelmingly report voxels with *increases* of BOLD signals for pain compared to no-pain stimuli, we therefore commit ourselves to a directional hypothesis and look for sites and neurons that increase their broadband activity or firing rate for stimuli that are perceived as more painful.

## Results

### The pain intensity ratings of patients were within the normal range

To assess whether the behavior of the patients was representative of the general population, we compared patients' (three males, four females, 34.3 years ± 9 SD) ratings with those of 93 healthy volunteers (54 females, 32.7 years ± 9 SD, Table 4), who took part in an online version of the video pain rating task. *Table 1* shows the distribution of ratings separately for patients and on average for controls. We calculated three metrics of similarity between the ratings of the patients and the control group: the Spearman's rank-order correlation, the slope, and the intercept of a simple linear regression between each patient's ratings and the average ratings of the control sample (*Figure 1e*). The patients revealed correlation coefficients, slopes, and intercepts (green and purple bars) within the 2.5 and 97.5 percentiles of the corresponding control sample distributions (gray bars), except for one correlation coefficient for Faces, where a patient rated the Faces with unusually high concordance with the average of the control groups. This verified that these seven patients were not impaired in their ability to rate intensity from our videos. In both the patient and the control sample, we observed that correlation coefficients for Faces were significantly greater than for Hands (patient: $t_{(6)}$ = 3.81,

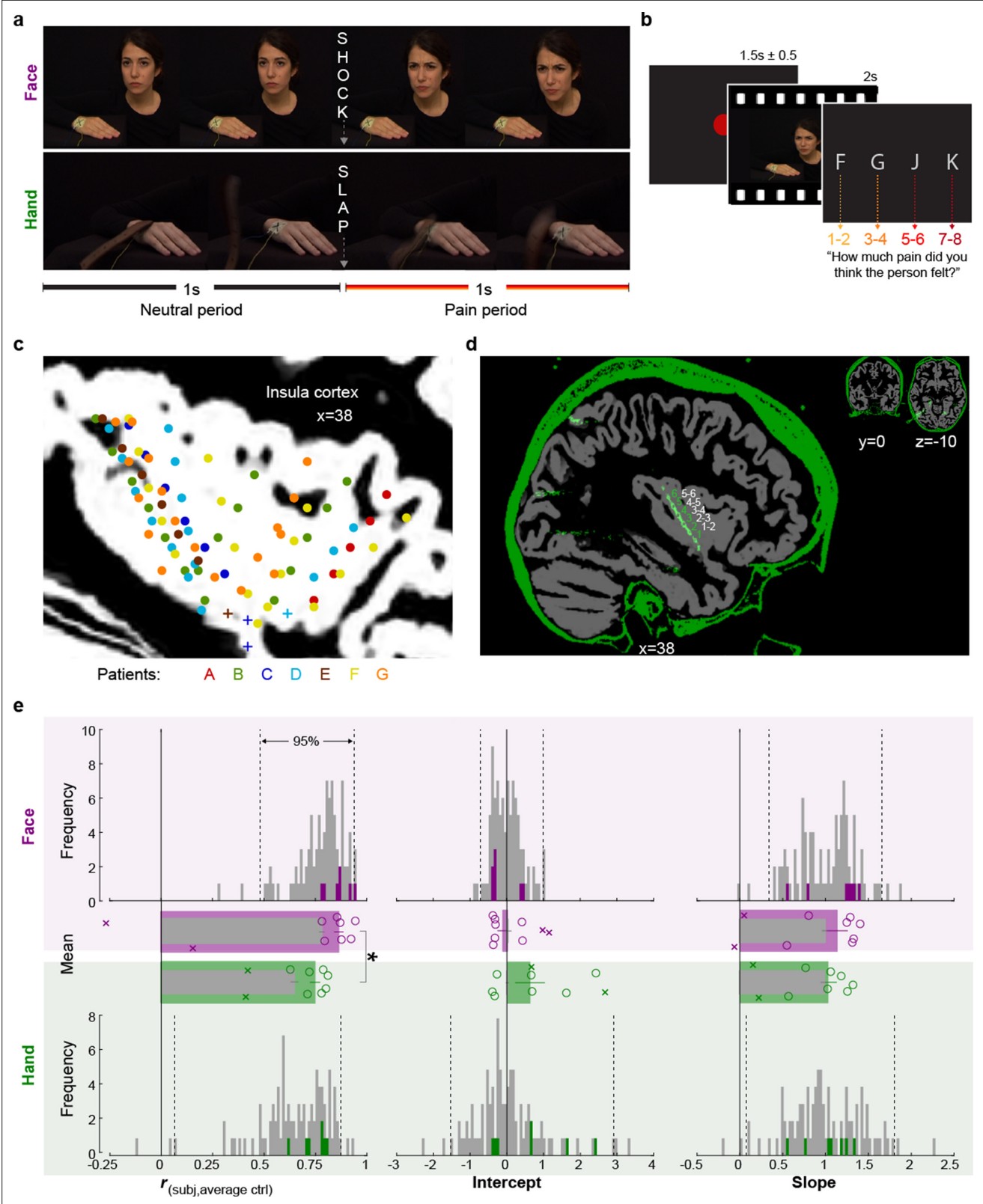

**Figure 1.** Experimental design, recording site locations, and behavioral pain ratings. (**a**) Frames extracted from a Hand and a Face movie. For the Face, the first second of each movie showed a neutral facial expression, the second, the facial reaction to the shock. For the Hand, the movie started with the belt resting on the hand. The first second showed the belt lifting and coming down again, to hit the hand at the 1 s mark exactly. The hand then reacted to the force of the belt. Both the slap and the shock delivery happened in the middle of the movies, splitting them into a 1 s neutral and a 1 s

*Figure 1 continued*

pain period. (**b**) Single-trial structure diagram. After the presentation of each video, patients expressed their choice at their pace using the keyboard keys f, g, j, and k for pain intensities 1–2, 3–4, 5–6, and 7–8, respectively. ITI started with participant's response. (**c**) Position (i.e., the midpoint between two adjacent electrodes) of the 85 bipolar macroelectrode recording sites shown as dots and of the microelectrode locations shown as pluses, color-coded by patient. Data from the two hemispheres and all lateromedial coordinates are projected here onto a single sagittal slice of the insula taken at X = 38 from the brain of one of the patients. For a list of all MNI coordinates, see *Figure 1—source data 1*. (**d**) Graphical illustration of how a bipolar recording for one patient and one insular electrode was computed. In green, the CT, and in gray, the T1 scan from patient C. The annular structures along the electrode shaft in the CT correspond to individual macroelectrode contacts (green 1, 2, 3 …). Recordings from adjacent pairs of contacts along the electrode were subtracted to calculate bipolar recordings (white 1–2, 2–3 …). (**e**) From left to right, Spearman's correlation coefficient *r*, intercept, and slope values from the linear regression for Hand (green) and Face (purple). Histograms: values for the control group illustrate the similarity between the ratings of each participant in the control group and the average of the other controls, and are shown as gray; the similarity between each of the seven patients with the average of the control group is shown in colors. Dotted lines mark the 2.5 and 97.5% of the control group. Bar graphs: mean ± SEM of the controls (gray) and the seven included patients (color) with individual patients as circles. In the bar graphs, we also show as Xs the corresponding behavioral performance metrics of the two patients that were excluded due to atypical use of the response keys. These patients were not included in the mean and SEM calculations.

The online version of this article includes the following source data for figure 1:

**Source data 1.** Mean MNI coordinates of recording sites.

$p_2$=0.009, $BF_{10}$ = 7.39; control: $W$ = 3895, $p_2$=$10^{-12}$, $BF_{10}$ = 3 × $10^5$, *Figure 1e*), suggesting more interpersonal agreement in pain intensity ratings for Faces than Hands. In contrast to the higher agreement for Faces, the average rating was slightly higher for Hands than Faces in both the patient and the control sample (patient: $t_{(6)}$ = 2.60, $p_2$=0.041, $BF_{10}$ = 2.31; control: $W$ = 2738.5, $p_2$=0.001, $BF_{10}$ = 39.40). Taken together, these findings indicate that the intensity rating behavior of the patient samples was similar to the patterns observed in the healthy population.

**Table 1.** Pain ratings in patients and controls.

Left: the percentage of trials (out of the 60 Hand and 60 Face trials for patients, or 30 and 30 for controls) per rating per participant for the Hand and Face conditions. For the age- and gender-matched control group, only the average across the 93 controls is shown. Middle: mean (M) rating for the Hand or Face. Our patients reported slightly higher pain intensity ratings for our Hand than Face stimuli ($t_{(6)}$ = 2.60, $p_2$=0.041, $BF_{10}$ = 2.31), the same was true for the age- and gender-matched controls (n = 93, $W$ = 2738.5, $p_2$=0.001, $BF_{10}$ = 39.40). This was somewhat surprising because the Hand and Face stimuli were rated as similarly intense in a validation study that preceded stimulus selection (*Gallo et al., 2018*). Right: standard deviation of the ratings for each participant. Because the efficiency of a regression depends on the standard deviation of the predictor, and much of our results depend on the relation between rating and intracranial electroencephalographic (iEEG) responses, we calculated the standard deviation for each participant and condition. The standard deviations were normally distributed (all Shapiro–Wilk p>0.25), we then used a *t*-test to compare them across the two conditions. We found no significant difference amongst the patients ($t_{(6)}$ = 1.44, $p_2$=0.199, $BF_{10}$ = 0.75). Differences we find in the correlations between rating and iEEG across Hand and Face stimuli therefore cannot be due to difference in the efficiency of these two estimations.

| | | Rating | | | | | | | | M (rating) | | SD (rating) | |
| --- | --- | --- | --- | --- | --- | --- | --- | --- | --- | --- | --- | --- | --- |
| | | Hand | | | | Face | | | | Hand | Face | Hand | Face |
| | | 1–2 | 3–4 | 5–6 | 7–8 | 1–2 | 3–4 | 5–6 | 7–8 | | | | |
| Patient | A | 1.67 | 38.33 | 43.33 | 16.67 | 40.00 | 28.33 | 25.00 | 6.67 | 2.75 | 1.98 | 0.75 | 0.97 |
| | B | 0.00 | 16.67 | 50.00 | 33.33 | 36.67 | 35.00 | 13.33 | 15.00 | 3.17 | 2.07 | 0.69 | 1.06 |
| | C | 0.00 | 0.00 | 43.33 | 56.67 | 56.67 | 43.33 | 0.00 | 0.00 | 3.57 | 1.43 | 0.50 | 0.50 |
| | D | 1.67 | 38.33 | 38.33 | 21.67 | 35.00 | 41.67 | 23.33 | 0.00 | 2.80 | 1.88 | 0.80 | 0.76 |
| | E | 28.33 | 28.33 | 31.67 | 11.67 | 35.00 | 20.00 | 28.33 | 16.67 | 2.27 | 2.27 | 1.01 | 1.12 |
| | F | 25.00 | 40.00 | 25.00 | 10.00 | 38.33 | 31.67 | 25.00 | 5.00 | 2.20 | 1.97 | 0.94 | 0.92 |
| | G | 38.33 | 23.33 | 26.67 | 11.67 | 36.67 | 28.33 | 25.00 | 10.00 | 2.12 | 2.08 | 1.06 | 1.01 |
| | Mean ± SEM | 13.57 ± 5.74 | 26.43 ± 5.10 | 36.90 ± 3.29 | 23.10 ± 5.90 | 39.76 ± 2.68 | 32.62 ± 2.85 | 20.00 ± 3.50 | 7.62 ± 2.33 | 2.70 | 1.95 | 0.82 | 0.91 |
| Control | Mean ± SEM | 33.01 ± 2.13 | 38.89 ± 1.30 | 21.25 ± 1.73 | 6.85 ± 1.12 | 44.55 ± 1.69 | 32.76 ± 1.05 | 16.99 ± 1.19 | 5.70 ± 0.93 | 2.02 | 1.84 | 0.74 | 0.81 |

## LFP activity in the insula correlates with the perceived intensity of the pain of others

Correlating power with reported pain intensity, irrespective of whether Hand or Face videos were shown, revealed a cluster of positive correlations ranging from 20 to 190 Hz and 1.12–1.62 s ($p_1<0.001$; $p_1$=one-tailed p-value), another cluster of positive correlations at very low frequencies (1–6 Hz, 0.02–2.06 s; $p_1<0.001$), and a small cluster of negative correlations (13–17 Hz, 1.30–1.83 s; $p_1=0.004$, not further discussed; *Figure 2a*, *Figure 2—figure supplement 1a*). Intensity coding was apparent in all traditional frequency ranges, except alpha (*Figure 2b*), and, as expected, was significant in the pain period. With no obvious differences among frequency bands above alpha, we henceforth used the frequency band 20–190 Hz for all analyses and refer to it as broadband power (BBP). We concentrate on BBP rather than oscillatory signals in lower frequencies because BBP is more closely linked to neural spiking (*Bartoli et al., 2019*; *Buzsáki et al., 2012*; *Miller et al., 2014*), cannot be explored in noninvasive EEG recordings, and is the frequency range that can supplement the information available for the substantial fMRI literature (*Boynton, 2011*; *Maier et al., 2008*). The temporal profile of the BBP–rating association revealed two periods with significant positive correlations: 1.14–1.54 s and 1.74–1.96 s (*Figure 2b*). Averaging BBP over the entire pain period revealed that, out of 85 macrocontacts within the insula, 27 (32%) showed a significant positive correlation (assessed as $p_1<0.05$, *Figure 2c*) between perceived intensity and BBP (n = 120 trials, all $r_{S(118)} > 0.156$, $p_1<0.045$), which was extremely unlikely to occur by chance (binomial $p_1=5 \times 10^{-15}$, $BF_{+0} = 3 \times 10^{12}$). Furthermore, randomly picking 85 electrodes anywhere in the brain (*Figure 2—figure supplements 2 and 3*) yielded BBP–rating associations that were significantly lower than those we found in the insula ($p_1=4 \times 10^{-5}$, *Figure 2d*), confirming that the BBP in the insula has enriched intensity coding. Splitting trials based on reported intensity and identifying moments in which the intensity coding is significant in an ANOVA confirmed that BBP scaled with pain ratings from 1.10 to 1.70 s (*Figure 2e*). Averaging the BBP over the 1 s neutral and 1 s pain period and using a period (neutral, pain) × rating repeated-measures ANOVA (rmANOVA) revealed a significant interaction effect ($F_{(2.445,205.348)} = 37.49$, $p=8 \times 10^{-17}$, $BF_{incl} = 85,925$, *Figure 2f*). Planned comparisons indicate that the effect of reported intensity depends mainly on BBP increases for the two highest intensity ratings relative to the neutral period and lower ratings (*Figure 2f*, *Table 2*).

## Intensity coding arises earlier for Hands than Faces

To investigate how intensity coding depends on the stimulus, we focused on the BBP range of interest (20–190 Hz), identified independently of stimulus type (*Figure 2a*), and found significant intensity coding for the Hand from 1.01 to 1.44 s (hereafter called early period) and for the Face from 1.75 to 1.86 s and from 1.91 to 1.98 s (jointly called late period, *Figure 2g*). The insula thus reflects, in broadband activity, the perceived intensity with differential time courses for the Hand and Face videos in this study.

To explore the shape of the BBP–rating relation, we averaged BBP over time for the early and the late periods for each pain rating separately (*Figure 2h*). For the early period, a stimulus (Hand, Face) × rating rmANOVA revealed a significant interaction (Greenhouse×Geisser-corrected $F_{(2.183,102.621)} = 13.55$, $p=3 \times 10^{-6}$, $BF_{incl} = 2 \times 10^6$). Planned comparisons provided evidence that BBP for Faces in the early period was similar for consecutively increasing painfulness level pairs, whereas an orderly increase in BBP was observed for increasing pain ratings for Hands from 3 to 4 onward (see *Table 3* for the results of the statistical tests). However, BBP for ratings of 1–2 was unexpectedly higher than ratings of 3–4. A similar ANOVA for the late period revealed evidence for the absence of an interaction ($F_{(3,141)} = 0.55$, $p=0.650$, $BF_{incl} = 0.03$). There was only a significant main effect of rating ($F_{(3,141)} = 16.54$, $p=3 \times 10^{-9}$, $BF_{incl} = 2 \times 10^7$), indicating that BBP in the late period of the Hand and Face videos together was the same for ratings 1–2 and 3–4, but thereafter showed significant increases with each consecutive increase in pain ratings (see *Table 3* for the results of the statistical tests). Taken together, these analyses indicate that BBP in the insula reflects perceived intensity only for the Hand stimuli in the early, and for both stimulus types in the late period.

## The timing of shape information matches that of Face intensity coding

Having observed differences in the temporal profiles of intensity coding for Hands and Faces, we next assessed whether these differences could arise from the timing of different intensity cues depicted

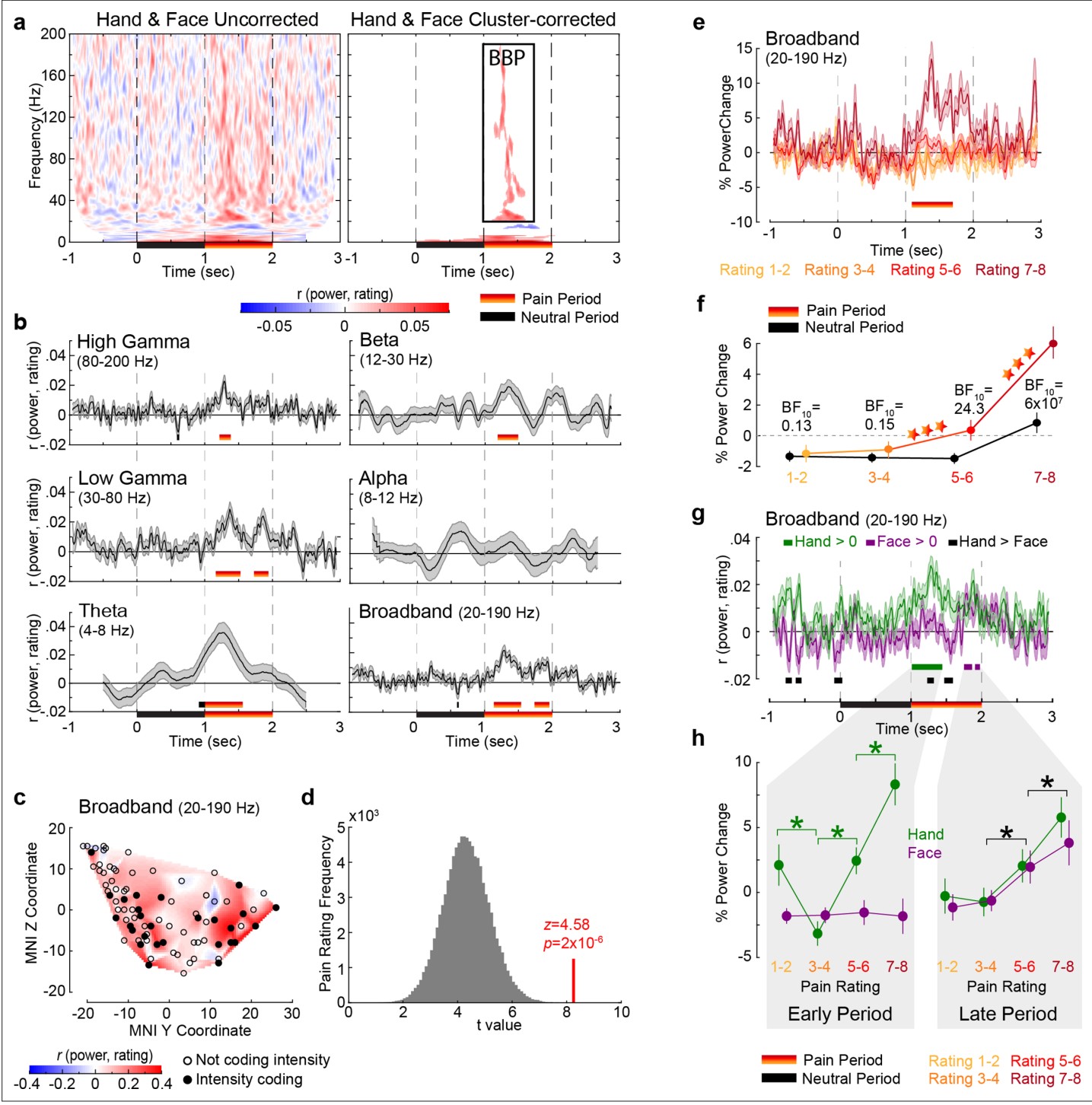

**Figure 2.** Intensity coding in the insula local field potential (LFP) activity for Hands and Faces together. (**a**) For each frequency and time relative to stimulus onset, the average $r_S$ value over all insular bipolar recordings between intracranial electroencephalographic (iEEG) power and rating for Face and Hand trials together, without (left) and with (right) cluster correction for multiple comparisons. BBP: broadband power, the cluster of significant positive intensity coding frequencies (i.e., $r_S > 0$, 20–190 Hz) used throughout the article. The time–frequency decomposition per rating can be found in *Figure 2—figure supplement 1*. (**b**) Mean ± SEM time course of intensity coding in different frequencies and BBP over all insular bipolar recordings when Face and Hand trials are combined. Above the x-axis, black and yellow-to-red bars show periods of significant intensity coding after circular shift correction for multiple comparisons during the neutral and pain periods, respectively. Below the x-axis, the black bar marks the neutral and the yellow-to-red bar indicates the pain period. (**c**) Intensity coding in the 85 bipolar recordings is shown as significant ($p_1 < 0.05$, filled black circles) or nonsignificant ($p_1 > 0.05$, open circles) based on the MNI y (anterior–posterior) and z (dorsoventral) coordinates. The heatmap shows the interpolated intensity coding

*Figure 2 continued on next page*

*Figure 2 continued*

values between these locations. Electrodes in the right and left insula are projected onto the same sagittal representation. (**d**) The *t*-value of a *t*-test comparing the intensity coding of all insular 85 bipolar recordings combining Hand and Face trials within the pain period (1–2 s post-stimulus onset) in the insula against zero (red bar) was higher than the distribution of the corresponding *t*-values obtained when performing the same test using 85 bipolar recordings randomly selected 100,000 times from the macroelectrode contacts of our seven patients anywhere in the brain (see ***Figure 2—figure supplement 2*** for a map of all macrocontacts and ***Figure 2—figure supplement 3*** for the anatomical distributions of these macrocontacts). (**e**) Mean ± SEM time course of percent power change from baseline in BBP (20–190 Hz) over all insular bipolar recordings and over the Hand and Face conditions, but plotted separately for trials rated 1–2, 3–4, 5–6, and 7–8. (**f**) Mean ± SEM percent power change values over all insular bipolar recordings as a function of reported intensity separately for the neutral (black) and pain (yellow-to-red) periods when combining Hand and Face trials. $BF_{10}$ values: Bayes factor quantifying evidence for $H_1$ relative to $H_0$ from a nonparametric *t*-test comparing BBP during the pain period against that during the neutral period. ***$p<0.001$ relative to the preceding reported intensity. See ***Table 2*** for a complete description of the statistical values. (**g**) Mean ± SEM time course of intensity coding in BBP (20–190 Hz) over all insular bipolar recordings for Hands and Faces separately. $r_S > 0$ indicated with green bars for Hands and purple bars for Faces. Black bars indicate $r_{S\_Hand} > r_{S\_Face}$. The early and late periods that result for Hands and Faces, respectively, are used throughout the article. ***Figure 2—figure supplement 4*** depicts the percent power change values as a function of time for ratings 1–2, 3–4, 5–6, and 7–8 separately for Hands and Faces. (**h**) Mean ± SEM percent power change in the broadband frequency over all insular bipolar recordings as a function of rating for Hands and Faces in the early and late periods separately. Green *$p<0.001$ for Hand. Black *$p<0.01$ for the main effect of rating, that is, combining Hand and Face. See ***Table 3*** for a complete description of the statistical values.

The online version of this article includes the following figure supplement(s) for figure 2:

**Figure supplement 1.** Time–frequency decomposition as a function of intensity rating.

**Figure supplement 2.** Glass brain representation of all macrocontacts available in the seven patients.

**Figure supplement 3.** Overview of the anatomical distribution of all macrocontacts available in the seven patients.

**Figure supplement 4.** Broadband power (BBP) time course as a function of rating and stimulus.

in the two video types. We first subjected our stimuli to more detailed, time-resolved analyses to describe the temporal evolution of the motion information in the Hand videos, and the motion and the shape information in the Face videos. Motion information for both Hands and Faces was quantified based on pixel-based intensity changes across consecutive frame pairs. Shape information for Faces was estimated using an automated face analysis software to extract the two most reliable shape features of painful facial expressions: how lowered the eyebrows and how tightened the eyelids are (facial action units [AUs] 4 and 7, respectively, ***Figure 3a***, ***Kunz et al., 2019***). Appendix 1 indicates that for Faces static shape information was sufficient to explain video ratings, while for Hands, the shape information was not.

Leveraging the high temporal resolution of our iEEG recordings, we next asked whether the motion or the shape information better matches the timing of our intensity coding for Faces in the insula. ***Figure 3a*** shows shape information increases toward the end of the movies with rating intensity. Comparing the timing of intensity coding for the Face in the insula BBP (purple bar in ***Figures 2g and 3a***) with the timing of the shape information for Faces (separation between the curves in ***Figure 3a***) shows a nice correspondence, with both BBP and shape information being highest

**Table 2.** Post-hoc comparisons of ***Figure 2f***.

To follow up on the repeated-measures ANOVA (rmANOVA) on the 1 s broadband power (BBP) with factors period (neutral, pain) × rating (1–2, 3–4, 5–6, 7–8), the table reports, for each contrast of interest indicated over the first two left columns: the average (SEM) of % power change, the *W* (if normality was violated) or *t* (when the data was normal) test values, and the two-tailed p and $BF_{10}$ values for the tested comparison.

| Period | Pain rating | % power change | W | $t_{(84)}$ | $p_2$ | $BF_{10}$ |
|---|---|---|---|---|---|---|
| Neutral vs. pain period | 1–2 | −1.34 (0.35) vs. −1.16 (0.56) | 1903 | | 0.742 | 0.13 |
| | 3–4 | −1.43 (0.32) vs. −0.92 (0.55) | 1801 | | 0.909 | 0.15 |
| | 5–6 | −1.48 (0.35) vs. 0.36 (0.67) | | 3.42 | 0.001 | 24.31 |
| | 7–8 | 0.84 (0.67) vs. 6.07 (1.05) | | 7.29 | $2 \times 10^{-10}$ | $6 \times 10^7$ |
| Pain period | 1–2 vs. 3–4 | −1.16 (0.56) vs. −0.92 (0.55) | 1966 | | 0.545 | 0.12 |
| | 3–4 vs. 5–6 | −0.92 (0.55) vs. 0.36 (0.67) | 1065 | | $8 \times 10^{-4}$ | 15.21 |
| | 5–6 vs. 7–8 | 0.36 (0.67) vs. 6.07 (1.05) | 309 | | $3 \times 10^{-11}$ | 950,944 |

**Table 3.** Post-hoc comparisons of *Figure 2h*.

To follow up on the two significant stimulus (Hand, Face) × rating repeated-measures ANOVAs (rmANOVAs), one for the early, one for the late period, the table reports, for each contrast of interest indicated over the first three left columns: the average (SEM) of % power change, the $W$ (if normality was violated) or $t$ (when the data was normal) test values, and the two-tailed p and $BF_{10}$ values for the tested comparison. The degrees of freedom were 47, as n = 48 since all possible rating options were only used by four patients with a total of 48 electrodes. Patients who used only some of the ratings are included in analyses using $r$(BBP,rating), but cannot be included in this rmANOVA approach.

| Period | Stimulus | Pain rating | % power change | $W$ | $t_{(84)}$ | $p_2$ | $BF_{10}$ |
|---|---|---|---|---|---|---|---|
| Early period | Hand | 1–2 vs. 3–4 | 2.11 (1.59) vs. –3.16 (0.94) | 1014 | | $3 \times 10^{-6}$ | 847.14 |
| | | 3–4 vs. 5–6 | –3.16 (0.94) vs. 2.44 (1.03) | | 5.97 | $3 \times 10^{-7}$ | 51,110 |
| | | 5–6 vs. 7–8 | 2.44 (1.03) vs. 8.32 (1.61) | 188 | | $2 \times 10^{-5}$ | 764.63 |
| | Face | 1–2 vs. 3–4 | –1.81 (0.6) vs. –1.75 (0.6) | | 0.1 | 0.92 | 0.16 |
| | | 3–4 vs. 5–6 | –1.75 (0.6) vs. –1.54 (0.95) | | 0.25 | 0.803 | 0.16 |
| | | 5–6 vs. 7–8 | –1.54 (0.95) vs. –1.83 (1.35) | | 0.23 | 0.817 | 0.16 |
| Late period | Hand and Face | 1–2 vs. 3–4 | –0.76 (1.01) vs. –0.73 (0.84) | 597 | | 0.931 | 0.16 |
| | | 3–4 vs. 5–6 | –0.73 (0.84) vs. 1.96 (1.13) | | 3.46 | 0.001 | 25.15 |
| | | 5–6 vs. 7–8 | 1.96 (1.13) vs. 4.75 (1.44) | | 2.9 | 0.006 | 6.29 |

toward the end of the movie. Furthermore, a partial least-squares regression (PLSR) analysis indicated that the time course of shape information could predict the rating of our patients with very high accuracy (*Figure 3d*). Regarding kinematics, we calculated the changes in pixel values across consecutive frames to track the timing of motion (*Figure 3b*), and this information could also predict the rating of our patients with high accuracy for Faces (*Figure 3e*). Comparing the timing of intensity coding in the insula for Faces (purple bar in *Figure 2g*) with the timing of motion information (separation between the curves in *Figure 3b*) shows that intensity coding maximizes when motion information has already declined significantly.

We complemented these observations with a quantitative approach that estimates how the neural intensity coding lags behind the shape or motion information (*Figure 3g, h, j and k*). If motion were, for instance, the driver of neural response to Face stimuli, we would expect that when motion information increases, neural responses should start increasing within ~200 ms, given typical latencies in the insula for facial expressions (*Chen et al., 2009*; *Cornwell et al., 2008*; *Krolak-Salmon et al., 2003*; *Meeren et al., 2013*) or other painful or nonpainful sensory stimulations (*Bastuji et al., 2018*; *Bastuji et al., 2018*; *Kobayakawa et al., 1996*; *Liberati et al., 2020*; *Liberati et al., 2016*; *Taniguchi et al., 2022*). Thus, we conducted correlation analyses to test how much the temporal profiles of shape or motion information are associated with the temporal profiles of intensity coding at various lags. Note that to directly contrast the predictive power of the shape and motion information, we used partial correlations. For Faces, partial correlations were positive for shape information in time windows centered at 40–320 ms and for motion at 560–1000 ms lags (*Figure 3j and k*). As discussed in more detail in the section 'Discussion,' given typical insular response onset latencies, intensity coding for Faces in the insula is more likely to be primarily triggered by shape information.

## Rating-related motion information could drive Hand intensity coding

Motion energy is also a reliable predictor of pain intensity ratings for Hands (*Figure 3f*). In our Hand videos, motion occurs at two time points: early, when the belt is lifted up, and then again, around the moment when the belt hits the hand (*Figure 3c*). Pain, however, occurs only at the second point in time. This allows us to explore whether the insula is coding movement in general, or movement that is associated with pain more specifically. We thus divided the 2 s of the Hand movies into six segments and asked, for each segment, how well BBP relates to motion energy in the same segment (*Figure 3m and n*). Over the 85 channels, we had evidence of absence for a relationship during the neutral period that contained the period during which the belt was seen to move upward (all $BF_{10} < 1/3$), and evidence for a relationship during the first 666 ms of the pain period when the belt is seen to slap the hand (both

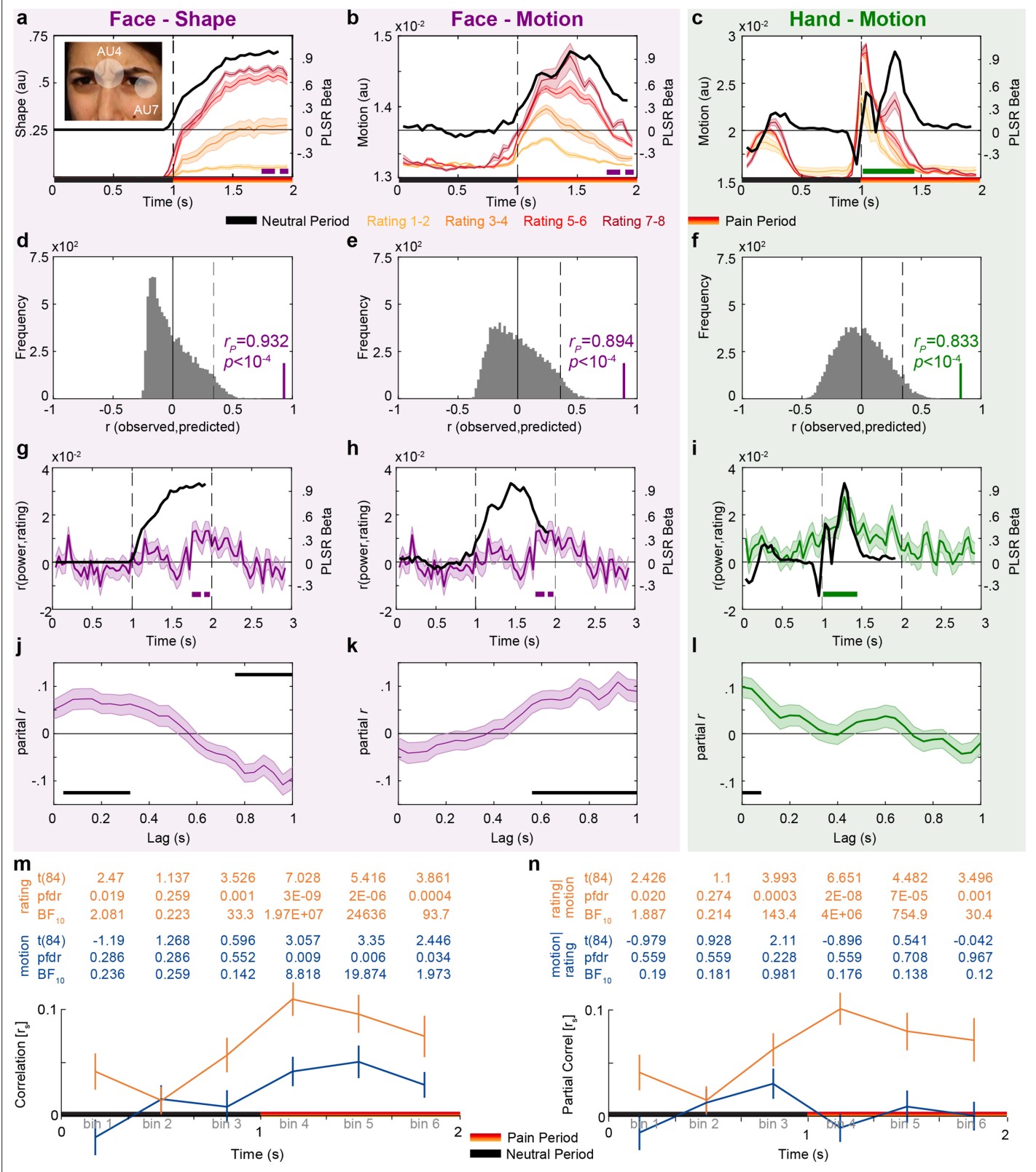

**Figure 3.** Temporal dynamics of pain rating and intensity coding in the insula broadband activity. (**a–c**) Motion and shape signals as a function of time and perceived intensity for the Face and Hand videos rated as 1–2, 3–4, 5–6, and 7–8 separately. Each colored curve represents the mean ± SEM for each rating. Purple and green bars indicate the periods with significant broadband power (BBP)–rating correlations for Faces and Hands, respectively (as in *Figure 2g*). Black lines represent the partial least-square regression (PLSR) beta coefficients predicting perceived intensity ratings using motion

*Figure 3 continued on next page*

*Figure 3 continued*

(for Hand and Faces) or shape information (for Faces). The white transparent circles over the inlet figure in (**a**) shows the action units (AUs) 4 and 7 that were used to estimate the intensity of the shape information in Face videos. (**d–f**) Accuracy with which the motion or shape signal across all frames can be used to predict the intensity rating of the movie. The histogram shows the actual predictive accuracy averaged over cross-validation folds (green and purple) relative to the null distribution of shuffled ratings (gray), with the median and top 5% of the null distribution shown as full and dashed line. In all cases, the actual accuracy was higher than all 10,000 shuffling values, as indicated by $p<10^{-4}$. (**g–i**) Mean ± SEM time courses of the correlations between BBP and pain ratings (green and purple, as in *Figure 2g*) superimposed with black lines from (**a–c**) for visualization of the temporal similarity between the two curves. (**j–l**) Mean ± SEM lagged correlation (left) and partial correlation coefficients (middle and right) between the temporal profile of BBP–rating correlations and that of the PLSR beta coefficients for the corresponding stimulus information. For partial correlation analyses, middle panel shows $r_P(BBP(t),Motion(t+lag)|Shape(t+lag))$ and the right panel shows $r_P(BBP(t),Shape(t+lag)|Motion(t+lag))$. All correlations are shown for lags from 0 to 1000 ms in steps of 40 ms. The correlation was calculated separately for each of the 85 bipolar recordings. The black bars represent periods of significant correlations, tested using a *t*-test of the 85 correlation values against zero followed by false discovery rate (FDR) correction at q = 0.05. (**m**) Mean ± SEM $r_s$ between motion energy and BBP (blue) or between subjective rating and BBP (orange) for the six consecutive bins of 333 ms during the movie. All statistics are two-tailed parametric *t*-tests against zero because $r_s$ values were normally distributed (all Shapiro–Wilk p>0.05). Values are indicated in the table above each panel for each time bin of ⅓ s. FDR correction is over the six bins. No $r_s$-to-z transform was used because the $r_s$ values were in the range $-0.5 < r_s < 0.5$ for which r and z values are extremely similar. (**n**) As in (**m**), but partial correlations: $r_s(BBP,motion|rating)$ in blue and $r_S(BBP,rating|motion)$ in orange.

$p_{unc}<0.003$ or $p_{FDR}<0.018$ corrected for six bins, both $BF_{10} > 8.8$). Indeed an rmANOVA comparing the correlation values across the six bins confirms that the relationship between motion energy and BBP changes as a function of time ($F_{(5,420)} = 2.9$, p=0.014, $BF_{incl} = 1.52$). This shows that the BBP response in the insula does not code motion in general, but motion at a time when it is relevant, here to assess the pain intensity. Next, we asked whether subjective rating or motion energy was the best predictor of BBP across the six bins (*Figure 3m and n*). Rating per se was an even better predictor of BBP than motion energy (rmANOVA, 2 predictor × 6 bin, main effect of predictor: $F_{(1,84)} = 23$, $p=7 \times 10^{-6}$, $BF_{incl} = 13,473$). Interestingly, using partial correlations, we see that the correlation between rating and BBP remains highly significant when seeing the belt hit the hand even after removing what can be explained by motion energy, but we have evidence for the absence of a correlation between motion energy and BBP if removing the variance explained by rating (*Figure 3n*). Together, this data supports the idea that the insula could employ motion to encode the painfulness in our Hand videos, but does not respond to simply seeing the motion of the belt, and that subjective rating of intensity appears to mediate the relationship between motion and insular response.

We also conducted the same lagged correlation analysis for Hands, as described for Faces above: that is, calculating the correlation coefficients between the temporal profile of the motion information and the temporal profile of intensity coding for Hands at various lags (*Figure 3i and l*). This analysis showed that intensity coding in the insula is best associated with motion information in Hands in time windows with center points that lag behind by 0–80ms (*Figure 3l*).

## The insula contains a surprising number of intensity coding locations with stimulus preference

We next focused on how individual recording sites in the insula reflected perceived intensity. In the early period, for Hands, 21/85 (25%) showed significant intensity coding (rating–BBP correlations, n = 60 trials, all $r_{s(58)}>0.219$, $p_1<0.046$), which was above chance (binomial, 21/85 at alpha = 0.05, $p_1=9 \times 10^{-10}$, $BF_{+0} = 2 \times 10^{7}$). In contrast, for Faces, only 3/85 (4%) showed intensity coding in the early period, which is expected by chance (binomial $p_1=0.804$, $BF_{+0} = 0.03$). During the late period, above-chance numbers of recordings showed intensity coding for Hands (14/85, 17%, $p_1=8 \times 10^{-5}$, $BF_{+0} = 201.41$), and the same was true for Faces (15/85, 18%, binomial $p_1=2 \times 10^{-5}$, $BF_{+0} = 808.49$; *Figure 4a*).

If the insula simply represents salience, one might expect a tight association between intensity coding for Hands and Faces, and an above-chance number of locations showing dual-intensity coding for both Faces and Hands. In contrast, if the insula also represents more specific information, we would expect above-chance numbers of locations with intensity coding for Faces, but not Hands and vice versa. Statistically, we infer the presence of intensity coding based on $r_s > 0$, $p_1<0.05$, like elsewhere in the article, and its absence using Bayesian statistics (*Keysers et al., 2020*), with $BF_{+0} < ⅓$. Plotting each bipolar recording's $r_s$ values on an x–y plot, with x representing $r_s$ for Hands and y for Faces, with dashed and dotted lines at critical $r_s$ values corresponding to $p_1<0.05$ and $BF_{+0} < ⅓$, we define nine quadrants, three of which are of conceptual importance (*Figure 4b*): those of locations

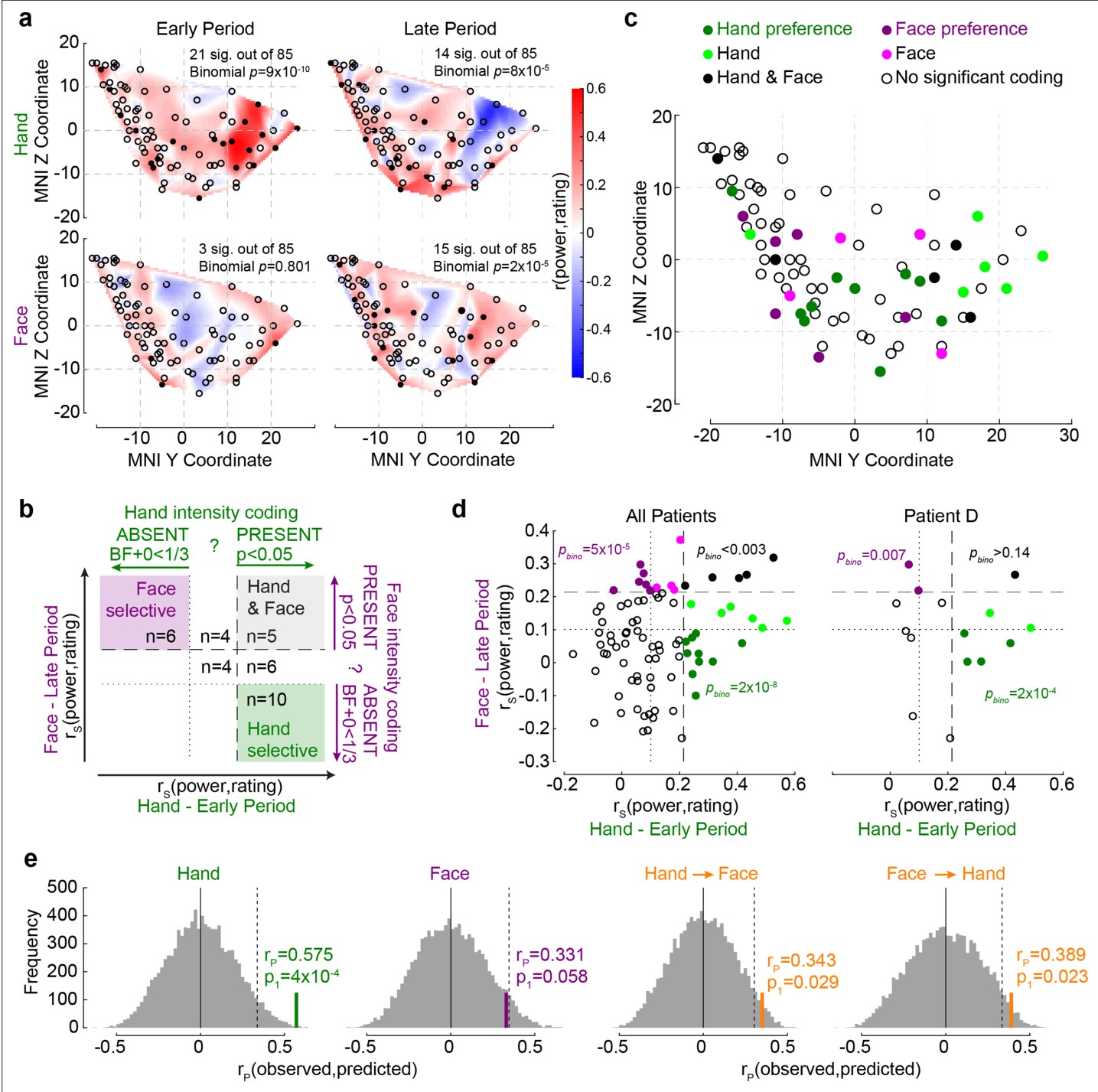

**Figure 4.** The relationship between Hand and Face intensity coding in the insula broadband activity. (**a**) Topographical maps of broadband power (BBP)–rating correlation coefficients for Hands and Faces in the early and late periods. Each circle is one of the recording sites (as in **Figure 1c**), with filled circles indicating locations with significant correlation coefficients ($p_1 < 0.05$). (**b**) Classification of recording locations based on their Hand (early period) and Face (late period) intensity coding. Bipolar recordings in the gray zone (n = 5) significantly co-represent intensity for Hands and Faces (both $p_1 < 0.05$, i.e., beyond dashed line). Recordings in the purple (n = 6) and green (n = 10) zone represent intensity coding preference for Faces or Hands, respectively (i.e., $p_1 < 0.05$ for Hands and $BF_{+0} < \frac{1}{3}$ for Faces, and vice versa). (**c**) Location of all 85 bipolar recordings, color-coded by stimulus preference as described in (**b**). Note that locations Hand and Face without further specification are those with $r_S$ values for at least one of the stimulus types falling between the dashed and dotted lines, thus providing inconclusive evidence and showing neither significant dual coding, nor evidence of absence. (**d**) Correlation coefficients for Hands and Faces separated by coding characteristics in (**b**) for all patients together (left) and for an exemplary patient (right). $p_{bino}$ refers to the likelihood to find the observed number of locations in that quadrant using a binomial distribution as detailed in section 'Probability

*Figure 4 continued on next page*

*Figure 4 continued*

of Face-but-not-Hand, Hand-but-not-Face, and dual-intensity coding LFPs'. (**e**) The left two panels depict the average correlation coefficients, together with corresponding resampling null distributions, as a measure of the accuracy of decoding intensity ratings using the partial least-square regression (PLSR) beta coefficients of BBP in the early period for Hands and in the late period for Faces. The right panels are similar to the left panels, but show the accuracy of cross-decoding, that is, predicting Hand ratings from the Face BBP and vice versa. The dotted lines indicate 95th percentiles of the resampling null distributions.

with dual-intensity coding (i.e., $p_1 < 0.05$ for Faces and Hands), those with intensity coding preference for Faces (i.e., $p_1 < 0.05$ for Faces, but $BF_{+0} < \frac{1}{3}$ for Hands) and those with intensity coding preference for Hands (i.e., $p_1 < 0.05$ for Hands, but $BF_{+0} < \frac{1}{3}$ for Faces). We then used binomial tests to compare the proportion of locations falling in these three quadrants against chance and found that all three quadrants contain more locations than expected by chance (*Figure 4d*). Indeed even within a single patient, among simultaneously recorded channels, we find above-chance numbers of channels with Face coding and with Hand coding preference (*Figure 4d*). Also, calculating the association between intensity coding across Hand and Face through a simple correlation of the respective *r* values confirms the presence of a significant but weak and barely worth mentioning (in a Bayesian sense) association ($r_K = 0.131$, $p_1 = 0.038$, $BF_{+0} = 1.27$). Together, this shows the insula is a patchwork, with some locations representing the Hand but not the Face, others the Face but not the Hand, and a small number finally representing both in terms of intensity coding. The spatial distribution of these locations is shown in *Figure 4c*.

In addition, we used a multivariate partial least-square regression (PLSR) approach to assess how well the pattern of BBP across the insula can predict participants' pain ratings. BBP across the 85 sites in the early period for Hands can be used to predict the patients' average rating of the stimulus with reasonably high accuracy (n = 10 trials since 1/3 of the 30 unique videos were used for testing decoding performance for each randomization, $r_{P(8)} = 0.575$, $p_1 = 9 \times 10^{-4}$ based on reshuffled distribution), and BBP in the late period for Faces with almost significant accuracy ($r_{P(8)} = 0.331$, $p_1 = 0.058$, *Figure 4e*). A direct comparison of the performance of the two PLSR indicates that the performance was higher for Hands than for Faces (nonparametric test across the decoding performances, $W = 944,605$, $p_2 = 9 \times 10^{-260}$, $BF_{10} = 7 \times 10^{39}$). To test whether intensity was encoded through similar patterns for the two stimulus types, we repeated the analyses training the PLSR on one stimulus type and testing it on the other. We found above-chance cross-decoding in both cases (Hand -> Face: $r_{P(8)} = 0.343$, $p_1 = 0.029$; Face -> Hand: $r_{P(8)} = 0.389$, $p_1 = 0.023$; *Figure 4e*). However, when the five contacts that significantly co-represented intensity for both Hands and Faces (black dots in *Figure 4c*) were excluded from the analyses, the cross-decoding accuracy fell to insignificant levels (Hand -> Face: $r_{P(8)} = 0.175$, $p_1 = 0.153$; Face -> Hand: $r_{P(8)} = 0.185$, $p_1 = 0.149$). These findings corroborate the above results, indicating that perceived intensity is reflected in the insula as a mixture of Hand-but-not-Face, Face-but-not-Hand, and Hand–Face common representations.

## Intensity coding for Hands increases anteriorly as in a similar fMRI experiment

To examine the spatial distribution of intensity coding, we examined the relationship between MNI coordinates of the bipolar recordings and intensity coding (i.e., $r_S$(BBP,rating), *Figure 5a*). The only significant association was that more anterior recordings (i.e., more positive y-coordinates) have higher Hand intensity coding. Interestingly, we found evidence against a right–left difference (i.e., $BF_{10} < \frac{1}{3}$ for x-coordinates) for the Face and Hand, providing moderate evidence against a left–right lateralization. To exclude that this finding could be driven by differences across patients, we also performed a random intercept mixed linear model using x, y, and z coordinates as predictors of Hand intensity coding (without interactions) with patients as random nesting variables. This analysis confirmed the y coordinates predict intensity coding for Hands (X: $F_{(1,79.23)} = 0.02$, $p_2 = 0.881$; Y: $F_{(1,80.97)} = 13.23$, $p_2 = 5 \times 10^{-4}$; Z: $F_{(1,73.95)} = 0.17$, $p_2 = 0.685$).

To better understand the origin of the anterior gradient for intensity coding for Hands, we performed a regression analysis between intensity coding of the 85 insular recording locations (for Hands and Faces separately) and resting state connectivity seeded at corresponding MNI locations in Neurosynth (*Yarkoni et al., 2011*). Insular locations with higher Hand intensity coding had higher resting state connectivity with the left anterior insula and ventral prefrontal cortex (including BA44/45,

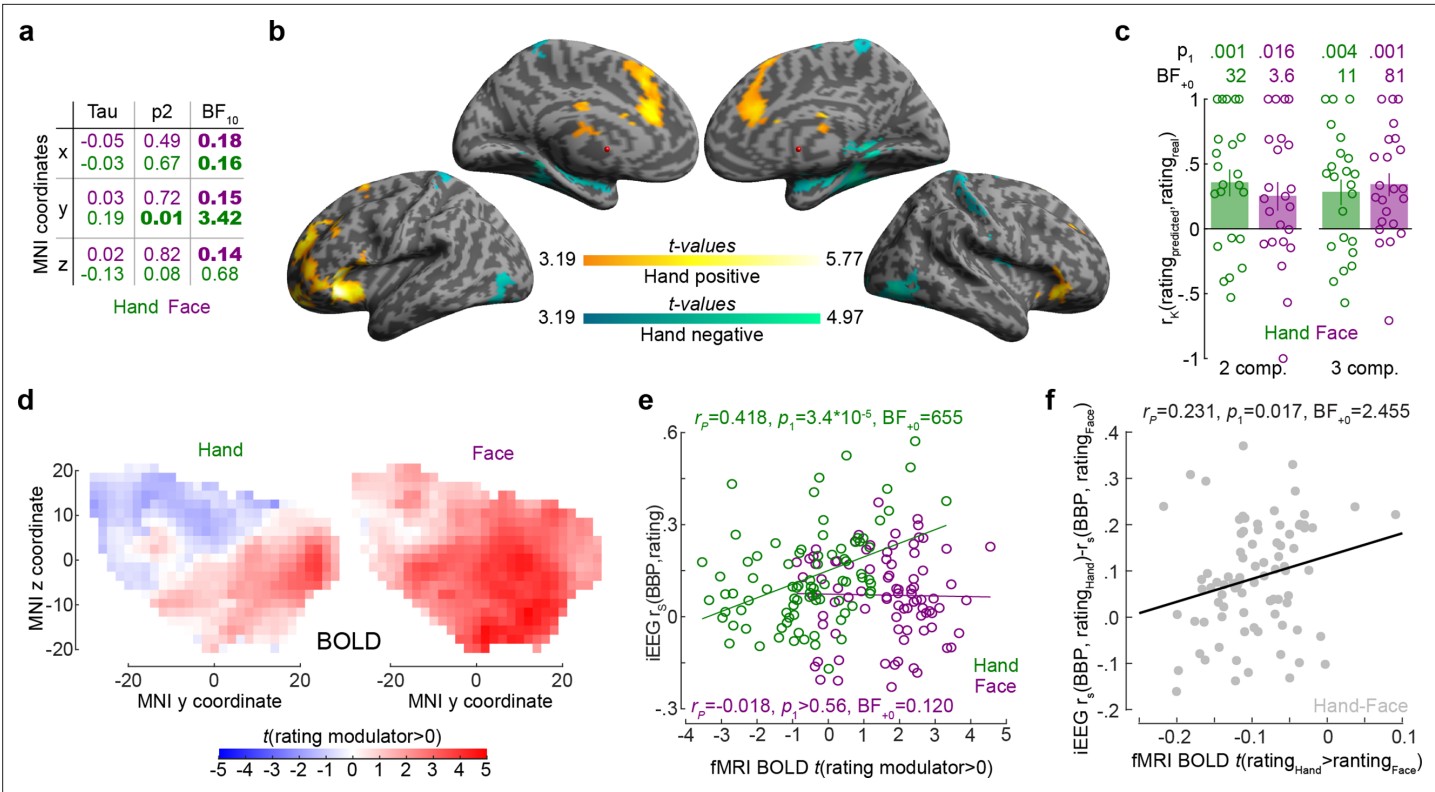

**Figure 5.** The relationship between the insula broadband and blood-oxygen-level-dependent (BOLD) activity during pain intensity ratings. (**a**) Correlations ($r_k$) between MNI coordinates and broadband power (BBP) intensity coding, separately for Hands (green) and Faces (purple). Bold numbers mark evidence for ($BF_{10} > 3$) or against ($BF_{10} < 1/3$) a significant correlation. Statistical values were obtained by correlating separately the x, y, or z coordinate of each bipolar recording with the $r_s$(BBP,rating) of each recording over all 85 recordings. Tau refers to Kendall's tau, $p_2$ and $BF_{10}$ the two-tailed probability and BF based on $H_0$:tau = 0. (**b**) Results of the regression analysis between resting state connectivity and intensity coding for the 85 bipolar recording coordinates for Hands. Significant positive and negative regression values are indicated by warm and cold colors, respectively. Results are corrected at the cluster level at $p_{FWE} < 0.05$ using initial cluster cutting at $p_{unc} < 0.001$, $t(82) = 3.19$, and then setting the minimum cluster size to FWEc = 772 as determined by the random field theoretical calculation in SPM. The detailed results of this analysis are provided in *Figure 5—source data 1*. (**c**) Mean ± SEM of the predictive performance of a partial least-square regression (PLSR) trained to predict ratings based on the pattern of BOLD activity across all voxels in the insula for different ratings. A leave-one-out cross-validation was used, and each circle represents the $r_k$ between the predicted and actual rating for each left-out participant, and the $p_1$ and $BF_{+0}$ values then test these n = 23 correlation values against zero using a nonparametric test. Results are shown separately for Hand and Face trials and using two or three PLSR components separately. (**d**) Topography of intensity coding for the Hand (left) and Face (right), as assessed at the group level, by the parametric modulator capturing changes in BOLD activity that correlate with trial-by-trial differences in participant's ratings. t-values testing the parametric modulator >0 at the group level are shown as a function of y and z coordinate in the insula mask. For each coordinate, the maximum value across all x coordinates within the two insulae is indicated. (**e**) Correlation ($r_P$ because of normality) between the t-value of the parametric modulator for the rating in the fMRI BOLD responses (x-axis) and the BBP intensity coding (computed in the early period for the Hand, green; late period for the Face, purple) in the intracranial electroencephalographic (iEEG) signal (y-axis) for each of the 85 contact locations. Note that for the fMRI signal the value is taken from the voxel closest to the MNI coordinates of the corresponding contact in the iEEG signal. (**f**) Same as (**e**) but for the difference between Hand and Face coding, calculated as the Hand–Face difference in the correlation between BBP and rating for the iEEG, and the t-value of the paired comparison between the parametric rating modulator for Hand–Face in the fMRI data.

The online version of this article includes the following source data for figure 5:

**Source data 1.** Resting state connectivity results.

OP8/9, Fp1), with the right frontal orbital cortex; with the bilateral cingulate (including BA24/33); and the right cerebellum (Crus I and lobules VI, VII, and VIII, *Figure 5b*, *Figure 5—source data 1*). In line with the lack of spatial gradients for Faces in the insula of our patients, examining which voxels had higher resting state connectivity with insular locations with higher Face intensity coding did not yield any significant voxels (all $p_{unc} > 0.001$).

Finally, we leveraged existing unpublished fMRI data from our lab to test whether the spatial gradient observed in the iEEG data resembles the spatial distribution of BOLD signals in the insula correlating with intensity ratings. Twenty-three independent healthy volunteers participated in the

fMRI-adapted version of the rating task. As for the iEEG experiment, participants were asked to rate Hand and Face 2 s videos on painfulness. The stimuli depicted the same actor as in the iEEG experiment and were made following the same procedure. To test whether the pattern of BOLD activity in the insula can be used to predict the ratings of the participants, we defined eight separate regressors in each participant, capturing all trials that participants rated with intensity 0–2, 3–4, 5–6, or 7–8 separately for Hand and Face trials. We then performed a PLSR with either two or three components using all voxels in the insula to predict the intensity rating. A leave-one-out cross-validation was used, and the correlation between the predicted and actual rating for each left-out participant was compared against zero. This confirmed that BOLD activity in the insula can be used to predict the perceived intensity for Hands (one-sample $t$-tests for two components: $t_{(22)} = 3.42$, $p_1 = 0.001$, $BF_{+0} = 32.32$ and three components: $t_{(22)} = 2.88$, $p_1 = 0.004$, $BF_{+0} = 10.95$) and Faces (one-sample $t$-tests for two components: $t_{(22)} = 2.78$, $p_1 = 0.016$, $BF_{+0} = 3.61$ and three components: $t_{(22)} = 3.86$, $p_1 < 0.001$, $BF_{+0} = 81.35$; *Figure 5c*), and performance did not differ across Hands and Faces (paired $t$-test comparing the leave-one-subject-out performance for Hands and Faces, with two components, $t_{(22)} = 0.675$, $p_2 = 0.507$, $BF_{10} = 0.27$; three components: $t_{(22)} = -0.39$, $p_2 = 0.700$, $BF_{10} = 0.23$). To compare the spatial pattern of intensity coding across the iEEG and fMRI data, we defined separate regressors for the Hand videos, Face videos, rating scale and button presses, and used the trial-by-trial ratings given by the participants as parametric modulators, one for the Face and one for the Hand trials, on the respective video regressor. For both Hands and Faces, visually inspecting *Figure 5d* reveals a gradient along the y-axis with more anterior locations showing stronger, and more positive associations between BOLD activity and rating. For Hands, across our 85 bipolar recordings in the patients, locations with higher BBP intensity coding in iEEG also show higher $t$-values comparing the parametric modulator for rating against zero in the BOLD signal (*Figure 5e*). For Faces, on the other hand, we found evidence of absence for an association of the two measures (*Figure 5e*). Finally, we also found that locations that had a stronger preference in their intensity rating for Hand over Face in the iEEG, also had a stronger preference in the fMRI data (*Figure 5f*).

## The insula contains neurons with intensity coding for Hands and/or Faces

The insula thus displays intensity coding in a broad frequency range, including locations with Hand-but-not-Face or Face-but-not-Hand intensity coding, as well as locations showing intensity coding for both stimulus types. To explore this representation at the level of single neurons, we analyzed the microelectrode data from the three patients (patients C, D, and E) that had microwires in the ventral anterior insula (pluses in *Figure 1c*). Spike sorting resulted in a total of 28 candidate neurons. From these, 13 showed more spikes during the pain period than the pre-stimulus baseline. Among those, eight show intensity coding for Faces and/or Hands (*Figure 6*), with significant Kendall's tau correlations between perceived intensity (1–2, 3–4, 5–6, 7–8) and spike count during the pain period (1–2 s post-stimulus onset) for at least one stimulus type: 4/8 for Faces and 5/8 for Hands (binomial test, Face: $p_1 = 0.003$, $BF_{+0} = 27$; Hands: $p_1 = 3 \times 10^{-4}$, $BF_{+0} = 282$). Considering the $p_1$-value for the intensity coding, two cells (a, b) showed intensity coding for both Hands and Faces, three (c–e) only for Hands, and three (f–h) only for Faces. If we additionally consider the $BF_{+0}$ values below ⅓ as evidence for the absence of coding in the other stimulus type, we find three Hand-but-not-Face coding cells (c–e) and two Face-but-not-Hand coding cells (g, h). Importantly, within patient D, we observe the coexistence of Hand-but-not-Face (c, d) and Face-but-not-Hand intensity coding (g).

To explore how spiking relates to BBP, we analyzed the BBP from the 10 microelectrodes that yielded the 13 cells showing stimulus triggered responses. Using Kendall's tau correlations between BBP (20–190 Hz) and patients' intensity ratings (1–2, 3–4, 5–6, 7–8) and comparing these results with the coding of the cells on the same wire reveals a relationship between the two. For Hands, 2/3 microelectrodes that yielded cells with intensity coding also showed significant association between ratings and BBP (*Figure 6a, c and d*). Indeed, intensity coding (i.e., correlation between intensity rating and spiking/BBP) was significantly correlated across the 10 microwires ($r_K = 0.57$, $p_1 = 0.012$, $BF_{+0} = 7.69$). For Faces, only 1/5 microelectrodes with spike intensity coding cells showed significant intensity coding in the BBP, and 2/5 showed a trend. Across the wires, there was a trend toward an association between the intensity coding in the spikes and BBP ($r_K = 0.34$, $p_1 = 0.088$, $BF_{+0} = 1.63$).

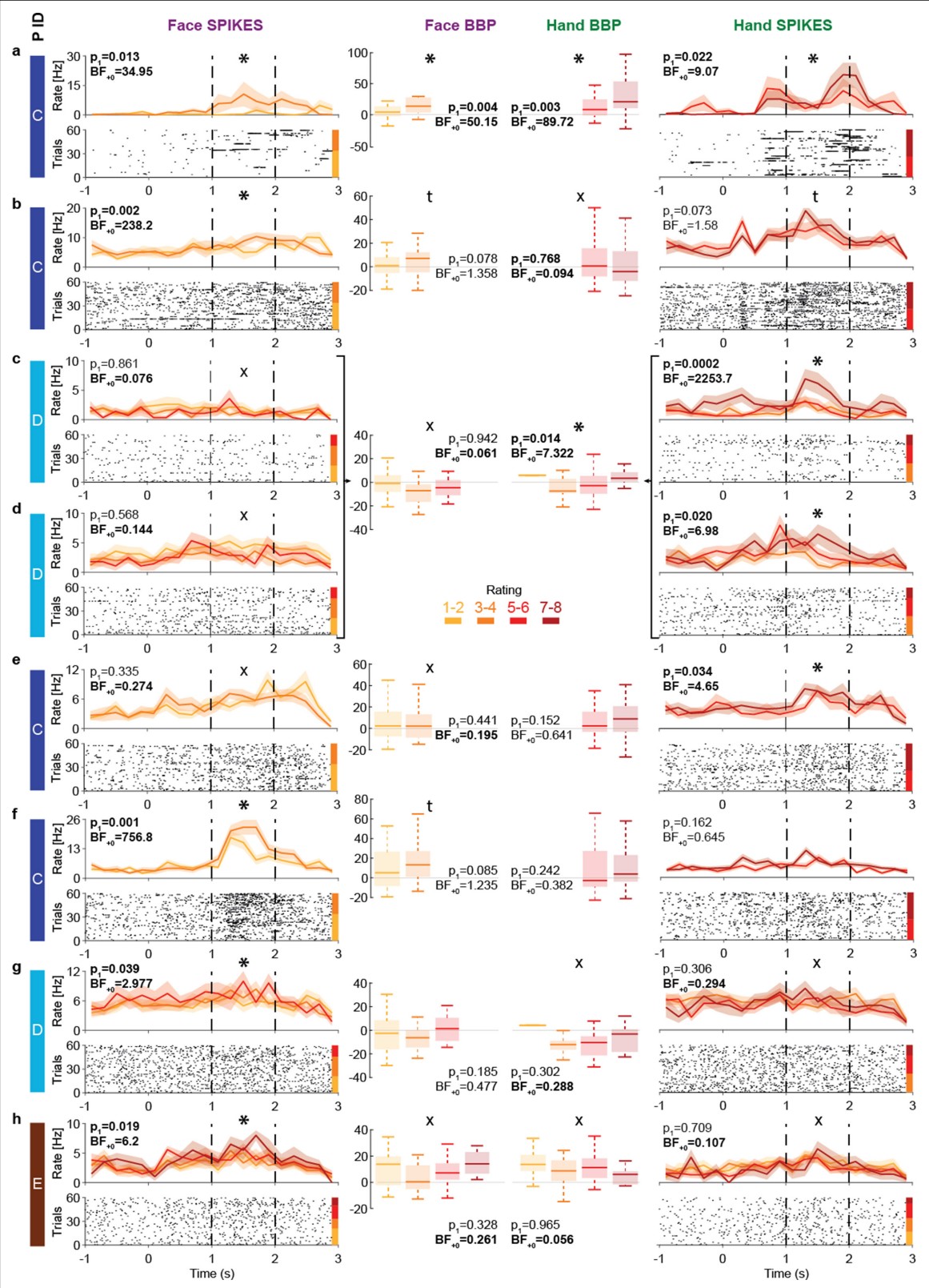

**Figure 6.** Intensity coding in the insula single units and the corresponding broadband activity. (**a–h**) Left (Face) and right (Hand) columns display, for each single unit, the rastergrams and peri-stimulus time histograms (PSTH) for eight cells that showed intensity coding for at least one stimulus type. For the PSTH, each curve represents the mean ± SEM of the firing rate in each bin for trials with the corresponding rating. Not all patients gave all possible ratings in each condition. For the rastergram, trials are sorted in order of rating, with the highest ratings upmost. The color bar next to the rastergram

*Figure 6 continued on next page*

*Figure 6 continued*

indicates the rows corresponding to each rating. $p_1$ and $BF_{+0}$ values result from a one-tailed test of the Kendall's tau between rating and spike count in the pain period (marked by the dashed lines). *Significant intensity coding ($p_1<0.05$), t: trend ($p_1<0.1$), X: evidence of absence for a positive intensity coding ($BF_{+0} < 1/3$). The x-axis (time) is relative to the movie onset. The color bar on the leftmost side indicates from which patient the data is taken. Middle columns show the broadband power (BBP) averaged over the pain period for the microelectrode (from which the corresponding single unit was extracted) as a function of rating where boxplots show variance across trials. The box and whiskers represent the quartiles across trials, and the $p_1$ and $BF_{+0}$, the Kendall's tau test of the association of rating and BBP. Note that cells c and d were taken from the same microwire, and therefore have only one BBP graph.

## Discussion

Here we characterize how the insula's iEEG activity encodes the intensity of other people's emotions using pain as an important category. LFPs indicate that neural activity in the insula within a broad range of frequencies, including the conventional theta, beta, and gamma frequency bands, scales with the perceived intensity of pain expressed by others. Interestingly, the insula only appeared to be recruited once the perceived pain level was at least moderate: activity was increased for moderate (5–6) compared to mild (3–4) and for severe (7–8) compared to moderate. However, activity for mild pain (3–4) was not significantly increased compared to minimal pain (1–2), or baseline activity. This echoes a recent finding that BBP activity in the insula is selectively increased only once thermal stimulation is consistently painful (*Liberati et al., 2020*). Furthermore, we isolate a small number of insular neurons increasing their firing with increases in the intensity of pain experienced by others.

As mentioned in the section 'Introduction,' BOLD signals can dissociate from neural spiking (*Boynton, 2011*; *Maier et al., 2008*). Just as V1 BOLD signals fluctuate with perception while spiking in V1 does not (*Maier et al., 2008*), the observation that BOLD signals in the insula fluctuate with perceived pain intensity alone cannot guarantee that neuronal spiking in the insula does. The insula's BOLD signal could instead fluctuate with perceived intensity simply as a result of feedback synaptic input from other brain regions that encode perceived intensity (e.g., area 24 in the cingulate gyrus; *Carrillo et al., 2019*). The foremost impact of our broadband gamma and spiking data is thus to provide what is arguably the first evidence that the intensity of other people's pain is indeed locally encoded in the activity of neurons in the insula.

The human insula has been in the focus of pain neuroscience as part of the pain matrix recruited by first-hand experience of pain (*Ingvar, 1999*). In this tradition, neuroimaging evidence for activation of the insula while witnessing pain experienced by others has led many to suggest it may represent a neural basis for empathy for pain (*Bernhardt and Singer, 2012*; *Jauniaux et al., 2019*; *Lamm et al., 2011*; *Timmers et al., 2018*). However, the insula is also recruited by a variety of tasks beyond nociception and pain empathy, including other affective states, sensorimotor functions, and decision-making under uncertainty (*Craig, 2002*; *Craig, 2009*; *Uddin, 2015*; *Uddin et al., 2017*). With so many tasks recruiting the insula, the idea has become popular that it may play a key role in attaching salience to behaviorally important stimuli (*Legrain et al., 2011*; *Uddin, 2015*; *Uddin et al., 2017*; *Zaki et al., 2016*). In this study, we do not intend to (and cannot) address the selectivity of the insula for the pain of others over other emotions, and we do not claim the neural responses we report are pain-specific. Instead, we characterize how the insula's iEEG activity encodes the intensity of other people's emotions, using pain as an important category, and use the agnostic terminology of 'intensity coding' rather than 'pain intensity coding' throughout the article to refer to recording sites. That the insula's broadband activity correlated with motion energy in our Hand stimuli only while the motion was associated with pain (during the slapping), but not while motion was innocuous (during the initial belt lifting), shows that the effects we measured cannot be reduced to visual motion detection. That perceived intensity was mediating the association between motion and broadband activity further speaks against an interpretation of our results as reflecting unspecific motion processing. Future experiments using a wider gamut of control stimuli that are matched in motion but differ in emotions will be critical to address the specificity of the responses we describe, and hence, what state they could reliably signal (*Zaki et al., 2016*). For the Hand stimuli, this could include seeing hands experience a range of different salient affective experiences, such as a hand being hurt, being caressed, and being rejected in addition to a neutral handshake (*Meffert et al., 2013*), or the actor in our Hand movies could have been asked to reposition their hand after each slap. For the face, this could include disgusted, angry, fearful, and happy facial expressions matched for motion. In general, using multiple

actors and painful hand interaction could help characterize how intensity coding in the insula is influenced by details of the stimuli that convey it, including the identity and gender of the actor, and how well our results generalize to other stimulus sets (*Yarkoni, 2020*).

An important and somewhat related question has been whether pain or salience cues are represented in a modality-specific or modality-general manner in the insula. fMRI studies have shown that the anterior insula is coactivated with different brain regions depending on whether the pain of others is conveyed via indirect cues or via the actual body part, such as the hand, that directly receives the noxious stimulation (*Gallo et al., 2018*; *Jauniaux et al., 2019*; *Keysers et al., 2010*; *Lamm et al., 2011*; *Timmers et al., 2018*). Here, we provide electrophysiological measures of neural activity that speak to that issue. We focus on the broadband gamma signal known to have comparatively high spatial specificity and be closest to neural spiking (*Bartoli et al., 2019*; *Buzsáki et al., 2012*; *Miller et al., 2014*), and find a mixed organization, consisting of modality-specific and -general locations in a partially intermixed layout. That is, we found locations with broadband activity and spiking associated with perceived intensity for the Hand, but not the Face; others associated with the Face, but not the Hand; and others still associated with perceived intensity for both. This echoes findings in recent fMRI studies that show that at the level of BOLD response within the insula some voxels encode intensity for both Hand and Face stimuli, while some only significantly contribute to intensity coding for one stimulus type (*Zhou et al., 2020*).

Leveraging our high temporal and spatial resolution, we found that locations showing intensity coding for the Hand stimuli have activity timing that echoes that of pain-related motion cues with very short lags (<80 ms). These optimal lags seem at odds with the much longer response onset latencies reported in the literature (*Bastuji et al., 2018*; *Bastuji et al., 2016*; *Chen et al., 2009*; *Cornwell et al., 2008*; *Krolak-Salmon et al., 2003*; *Liberati et al., 2020*; *Liberati et al., 2016*), but differ from response onset latencies in two ways. First, our BBP is calculated within time windows of 8× the inverse of the frequency. Hence, for the lowest broadband frequency of 20 Hz, power at time $t$ includes signals that occur in a window starting $4 \times 1/20 = 200$ ms earlier and ending 200 ms later. The peak of BBP would thus still align with the peak of neural activity, but the onset of our BBP estimate may underestimate the onset of neural activity by up to 200 ms. Second, *Figure 3i* illustrates that our short lags do not reflect a mysterious lack of delay between the *onset* of motion information and the *onset* of its neural encoding, but rather, the close alignment of their *peaks*. The onset of motion information (~920 ms) actually precedes the peak neural intensity encoding (~1300 ms) by almost 400 ms, in line with typical response latencies. The close alignment of stimulus information and neural encoding peaks rather reflects a property of naturalistic dynamic stimuli in which information typically builds up progressively: in such situations, neural processing, perhaps through predictive coding, appears to overcome significant neural response latencies to synchronize its peak responses to the peak of the stimuli (*Perrett et al., 2009*; *Reddy et al., 2015*). Locations that show intensity coding for the Face appear to have activity echoing the timing of shape information, with lags in the 40–320 ms range. These lags are in a range similar to those found in other studies using facial expressions in the insula (*Chen et al., 2009*; *Cornwell et al., 2008*; *Krolak-Salmon et al., 2003*; *Meeren et al., 2013*). Using automated software to detect the level of activation of the facial action units 4 and 7 (i.e., lowering the eyebrows and tightening the eyelids), we find that this action unit information suffices to predict participants' rating of the stimuli with high accuracy and follows the time course of the neural activity in the Face intensity encoding locations well enough to suggest that it provides traction on the analyses of dynamic pain-related facial expressions. On the other hand, the long neuronal lags we revealed for the facial motion information, with the earliest significant associations occurring in windows centered on 560 ms after the motion information, are unusually long for the start of responses to sensory stimuli in the insula as the onset latency for visual, auditory, tactile, or nociceptive responses is usually well below 560 ms (*Bastuji et al., 2018*; *Bastuji et al., 2016*; *Chen et al., 2009*; *Cornwell et al., 2008*; *Krolak-Salmon et al., 2003*; *Liberati et al., 2020*; *Liberati et al., 2016*), although insular activity can persist into such longer intervals (*Meeren et al., 2013*; *Taniguchi et al., 2022*). Together, this suggests that shape information is more likely than motion information to be the primary driver of the intensity coding to our facial stimuli.

An important consideration is whether preference for Face or Hand stimuli in specific locations could simply originate from some participants finding our facial stimuli more salient, and others the Hand stimuli, particularly given that our patients rated the Hand stimuli with slightly higher pain

intensity than our Face stimuli. That we find Face and Hand preference side-by-side in simultaneously recorded locations and neurons in single patients suggests that this cannot suffice to explain our data. This is because if a patient were to find Hand stimuli more salient than Face stimuli, and the insula simply codes saliency, we would expect to find Hand-but-not-Face intensity coding in that patient's insula, but we would not expect to find side-by-side locations with Hand-but-not-Face and Face-but-not-Hand coding. This also makes it unlikely that broadband activity simply reflects categorization uncertainty (*Grinband et al., 2006*). Indeed, it would be uneconomical for all insular neurons to redundantly encode the same exact signal of salience or categorization uncertainty. Our findings instead suggest that different locations might encode behaviorally relevant (and hence salient) information about the intensity of the pain of others with some degree of specialization for a particular type of information (Hand vs. Face). The insula intensity coding we measure could have a dual function: help perceive the intensity of other people's emotions and tag intense emotions as salient, thereby reconciling the notion that it contributes to empathy and saliency. One peculiar observation in our data is that BBP for the Hand stimuli did not increase monotonically as a function of reported pain intensity, but in a J shape, with higher power for the lowest than second lowest rating. Some have argued that the insula may be part of an active inference circuitry in the brain that learns to predict emotionally relevant states and stimuli (*Seth and Friston, 2016*). In such a perspective, a J-shaped response curve could reflect a combination of underlying neurons representing the intensity of other people's emotions (with a monotonic increase in activity) and others representing the prediction error (which for randomly presented intensities would have a U shape centered on the average intensity). Future experiments could contrast responses to a given pain intensity in blocks of high and blocks of low average presented intensity to disentangle the effect of observed and expected intensity on neural responses in the insula and shed further light on this predictive component.

On the other hand, we also found evidence that some locations and cells represent intensity coding for both the Hand and the Face stimuli. In addition, if we train a PLSR to predict perceived intensity from the activity pattern across all recorded locations, we find that training the decoder on Hand activity pattern and testing it on Face activity patterns (or vice versa) leads to above-chance decoding. This confirms that the insular representation of intensity can support stimulus-independent decoding – despite our PLSR not being biased to focus on signals that do generalize well across stimulus types. This provides an electrophysiological basis for recent fMRI studies that show that stimuli depicting situations in which others' experience is painful or not can be discriminated using the same pattern across Hand and Face stimuli (*Zhou et al., 2020*).

In addition to the broadband results we report in detail, we find that theta power increases with perceived intensity. Given a growing animal literature establishing that interareal theta synchronization promotes learning about threats (*Likhtik et al., 2014*; *Likhtik and Gordon, 2014*; *Taub et al., 2018*; *Tovote et al., 2015*), examining the coherence in the theta range across iEEG electrodes in different brain regions during pain observation may in the future shed light on how humans learn about safety through others.

Spatially, we found that Hand intensity coding was enriched in the anterior dorsal insula, where we also found the largest proportion of locations encoding both Hand and Face intensity. This anterior bias was also observed in our BOLD signal for similar Hand stimuli. A recent meta-analysis identified that the most consistent BOLD activations when observing limbs in painful situations within the insula occur bilaterally around MNI coordinates y = 13 and z = 10 (*Jauniaux et al., 2019*), which closely matches where we find the highest density of Hand intensity coding (*Figure 4a*). Interestingly, locations with higher Hand intensity coding have increased connectivity at rest with extrainsular regions involved in processing two relevant stimulus dimensions. Connectivity was higher with cerebellar lobules VI, VII, and VIII, and with the inferior frontal gyrus, all of which are recruited by *Abdelgabar et al., 2019*; *Caspers et al., 2010*; *Gazzola and Keysers, 2009* and necessary for (*Abdelgabar et al., 2019*; *Keysers et al., 2018*; *Pobric and Hamilton, 2006*) perceiving the very kinematics of hand actions we find to be good predictors of BBP activity in this study. Connectivity is also higher with the mid- and anterior cingulate cortex, which is associated with pain witnessing in humans (*Jauniaux et al., 2019*; *Lamm et al., 2011*; *Timmers et al., 2018*) and contains neurons with pain intensity coding in rats witnessing the pain of other rats (*Carrillo et al., 2019*).

With respect to Faces, somewhat surprisingly, we did not find a clear spatial clustering of intensity coding in our electrophysiological data. Overall, our ability to decode perceived intensity from our

Face stimuli was also lower than that from the Hand stimuli. In that context, it is important to reflect on the fact that, despite our efforts to match the perceived intensity based on previous data (*Gallo et al., 2018*), patients (and to a lesser extent an age- and gender-matched control group) perceived the Hand stimuli as more intense than the Face stimuli. Given that responses were strongest for the highest rating, and that this rating was given more rarely for Faces than Hands, this difference could have contributed to making our Face results less consistent. At the same time, the variance in rating, a critical factor for determining the efficiency with which a regression can detect the presence of a rating–BBP relationship, did not differ across Hand and Face. Together, this difference in perceived intensity cautions us against overinterpreting the lack of detectability of a topographic organization of responses to Faces. Indeed, in our BOLD data, where decoding performance was similar for Hand and Face stimuli, Face intensity coding also had a clear spatial organization, being stronger more anteriorly, and meta-analyses show the left anterior insula to be reliably recruited by the observation of painful facial expressions (*Jauniaux et al., 2019*). However, that we found fewer locations and less reliable spatial organization for Face than Hand intensity coding dovetails with recent meta-analyses of the fMRI literature showing that the insula is more reliably recruited by the sight of limbs than by painful facial expressions (*Jauniaux et al., 2019*; *Timmers et al., 2018*). Indeed, that we find a macroscopic organization of intensity coding for the Hand, but not the Face, is echoed at the mesoscale: microwires with cells with Hand intensity coding also tend to show Hand intensity coding in the BBP signal that is thought to pool the spiking of many neighboring neurons, but the same is not true for the Face. In terms of lateralization, we find that our data is more likely if one assumes that both hemispheres have similar intensity coding than if one hemisphere were dominant. This echoes the fact that during noxious stimulation of the right hand both insulae show significant iEEG responses (although slightly stronger in the left insula; *Liberati et al., 2020*), and that fMRI fails to find robust lateralization of responses to empathy for pain (*Jauniaux et al., 2019*; *Timmers et al., 2018*).

## Materials and methods

### iEEG experiment

#### Participants

##### Patients

Depth electrode recordings were collected from nine epileptic volunteers admitted at the Amsterdam UMC to localize seizure origin. Patients received initial study information from the neurosurgeon and provided informed consent to the experimenter before the surgery occurred. Our single-session experiment started on average 4 days after surgery (SD = 1.89 days). Preliminary analyses indicated that the pain rating performance of two patients (Xs in *Figure 1e*) for Face videos was significantly poorer compared to an age- and gender-matched healthy control group. Hence, these two patients were excluded from all analyses, which yielded a final sample of seven patients (four females, 34.3 years ± 9 SD, *Table 4*). The study was approved by the medical ethical committee of the Vrije University Medical Center (protocol 2016/037), and each patient signed a written informed consent according to the Declaration of Helsinki.

**Table 4.** Participants' demographics and epileptic status.
Our seven patients were matched in age and gender to the online control group from which we obtained normative movie ratings. The last column indicates the postoperative status of our patients. Three patients had other brain regions than insula surgically removed, and afterward had no more attacks (marked with 1), suggesting that the foci were clearly outside the insula. One patient had a region other than the insula surgically removed because the monitoring had suggested that the foci was outside of the insula; however, the patient continued to have post-surgical attacks (marked with 2). For three patients, no surgery was performed because there was no clear link between electrode locations and epileptic attacks (marked with 3).

| Group | Sample size (n) | Age (mean ± SD years) | Age-matched | Gender (M/F) | Gender-matched | Epileptic location score |
|---|---|---|---|---|---|---|
| Patient | 7 | 34.3 ± 9 | Mann–Whitney $U$-test, p=0.7, $BF_{01}$ = 3.6 | 3/4 | Multinomial test, p=0.7, $BF_{01}$ = 7 .3 | A = 1, B = 1, C = 3, D = 3, E = 1, F = 3, G = 2 |
| Control | 93 | 33.7 ± 9 | | 38/55 | | N/A |

Clinical investigation revealed that for all our patients the epileptic incidents did not appear to originate around the electrode contacts in the insula that we analyzed here. In addition, for four of them, recordings pointed to origins of the epilepsy to be clearly outside the insula, leading to the surgical removal of extrainsular regions (*Table 4*). Finally, for the remaining three, no clear origin for the epilepsy could be localized, but there was no indication that the insula was involved in the initiation of the attacks.

### Control participants in the online video rating task

To assess whether the behavior of the patients was representative of the general population, we compared patients' ratings with those of 93 volunteers (54 females, 32.7 years ± 9 SD), who took part in an online version of the video pain rating task. The matching with the seven patients was done by only including age- and gender-matched Dutch participants, and was successful (*Table 4*). The study was approved by the local ethical committee of the University of Amsterdam (2021-EXT-13608), and each participant signed an online informed consent form to participate in the study.

### Control participants in the online frame rating task

To determine whether participants could use shape information available in single frames to determine pain intensity in Hand and/or Face stimuli, 40 volunteers (23 females, 33.7 years ± 9 SD) from the same group who also performed the online video pain rating task participated in the online frame rating task, so they were already familiar with the videos and had a better understanding of where the single frames came from. This also allowed us to directly compare how they rate single frames with how they rated the movies from which the frames were taken. They were selected to approximate the age and gender distribution of the patient group.

## Stimuli and procedure

### Video rating task

The 2 s videos were generated as in *Gallo et al., 2018* and showed a Caucasian female receiving either electrical shocks to the hand (reaction conveyed by the facial expression only; Face condition) or a slap with a belt to the hand (reaction conveyed by the hand only; Hand condition). Hence, the location of the noxious stimulation was maintained across conditions (dorsum of the left hand), but the cues through which participants could deduce the painfulness differed. All videos started with 1 s of baseline: neutral facial expression for Face and static hand for Hand stimuli. The reactions of the actor were genuine, but enhanced: truly noxious stimuli were applied during recording, but the actor was encouraged to enhance her expressivity and not to suppress her reactions. This instruction was given in order to compensate for the fact that the intensity of the shock delivered had to be in balance with the fact that many videos had to be recorded. A mild pain was therefore used, in agreement with the actress, and enhancement was necessary to fully convey expressions of stronger pain. Movies were cut, so that evidence of pain started at 1 s (*Figure 1a*). Before the experiment, participants were instructed to rate pain intensity ('How much pain do you think the person felt?') on a scale from 0 (no pain at all) to 10 (the worst imaginable pain). To reassure patients that no real harm was inflicted to the actor in the movie, they were informed that during video recording stimulations in the 9–10 range were never used. Participants had to rate pain intensity after each video at their own pace, using four keyboard keys (*Figure 1b*). Only the relevant keys were presented on the screen, intensities were not indicated. Patients watched each of the 60 videos (30 Hand, 30 Face) twice in fully randomized fashion with a random interval of 1.5 s ± 0.5. The videos were matched in terms of intensity and standard deviation based on a validation in *Gallo et al., 2018*.

### Online video rating task

The stimuli and the task were the same as in the electrophysiology experiment, except each video was presented only once. Prolific Academic (https://www.prolific.co/) was used to recruit participants, and the experiment was implemented on Gorilla (*Anwyl-Irvine et al., 2020*; https://gorilla.sc/).

### Online frame rating task

The task was similar to the pain rating experiment, except still frames instead of the full videos were presented for 2 s. For faces, frames at the 1.8 s of the Face videos were used (except for one video

where the eyes were closed at 1.8 s, so the frame was taken at 1.68 s). This timepoint was selected because facial expressions were most pronounced toward the end of the movies, and more formal analyses confirmed that this corresponds to a time where shape information plateaued (*Figure 3a*). To use a comparable stimulus set for Hands, which portrayed maximal configuration information, we selected from each Hand video separately, the frame at which the hand was maximally depressed by the force of the belt slap (timepoint mean ± SD = 1.001 ± 0.013 s).

## Data acquisition

Patients were implanted with Behnke–Fried depth electrodes (Ad-Tech Medical Instrument Corporation; *Fried et al., 1999*) targeted at the right or left, anterior or posterior insula. Electrodes were inserted via a guide tube under the guidance of an online stereotactic positioning system. They consisted of a silastic hollow tube with 9–12 platinum outer macrocontacts, 1.28 mm in diameter, 1.57 mm in length with the first two macrocontacts spaced 3 mm from each other and the rest spaced 5 mm from each other. This hollow tube had nine platinum microwires (eight recording and one reference contact) running through it, each 38 micron in diameter, protruding as a 'pigtail' formation out of the tip of the electrode. Macrocontact recordings were amplified using unity gain, DC amplifiers (Braintronics BrainBox 1166 system), low-pass filtered at 1500 Hz (–3 dB point, –12 db/octave), and sampled at 32,768 Hz. The digital signal was decimated to a final sample rate of 512 Hz or 1024 Hz and was pre-filtered with a three-section FIR equiripple filter (0.01 dB passband ripple) with the passband set to 1/3 of the sample frequency and the stopband set to 1/2 of the sample frequency. Signals from the microcontacts were amplified with respect to a skull-screw ground using a unity gain HS-9 head-stage amplifier (NeuraLynx). The signal was high-pass filtered at 1 Hz and low-pass filtered at 5 kHz and had a sampling rate of 32 kHz. There were a total of 85 macroelectrodes and 32 microwires across all patients in the insula that we recorded from.

## Electrode localization

For each patient, the T1 structural MR image taken before the electrode implantation surgery and the CT scan taken after the electrode placement were co-registered (*Figure 1d*). Using SPM12 (https://www.fil.ion.ucl.ac.uk), the T1 image was segmented to determine the normalization parameters, and MR and CT images were then normalized to the MNI space using these parameters. CT scan and gray matter were overlaid with insula probability maps (*Faillenot et al., 2017*), and macrocontacts within the boundaries of the insula map were detected based on detailed visual examination using MRIcron (https://www.nitrc.org/projects/mricron). Not the individual subdivision maps, but the combination of all the subdivision maps as one general insula map was used for localization because the coverage of different subdivisions was highly uneven across patients. Since macrocontact recordings were analyzed in a bipolar layout, the coordinates of each bipolar recording were estimated as the midpoint of its macrocontacts (see *Figure 1—source data 1* for MNI coordinates).

## Data analysis

### General statistical approach

Much of the analyses in this article assess intensity coding, which examines the relationship between brain activity (measured based on LFP, spiking, or BOLD activity) and rating. Because the rating of pain intensity was along discrete categories (1–2, 3–4, 5–6, 7–8), which might be linear but is certainly ordinal, we tend to use association measures that are appropriate for ordinal scales when we relate brain activity to the rating of a single participant. That includes Spearman's $r$ in most of our MATLAB codes, when analyses need to be repeated for every electrode because it is the most widely used rank-order correlation metric. We use Kendall's tau when using Bayesian analyses implemented in JASP because these analyses are not yet available for Spearman's $r$. When examining the association between variables that are more continuous and normally distributed, we use Pearson's $r$.

When using $t$-tests, we examined normality using the Shapiro–Wilk test. If normality is preserved, we report $t$-tests and $t$-values; if not, we use Wilcoxon signed-rank or Mann–Whitney $U$-tests, as indicated by $W$ or $U$ values, respectively. When possible, or when evidence of absence is important for the interpretation of the data, we supplement the frequentist p-values with Bayesian statistics calculated using JASP (https://jasp-stats.org). We use the abbreviation $p_1$ to represent one-tailed p-values and

$p_2$ to represent two-tailed p-values. $BF_{10}$ and $BF_{01}$ represent relative evidence in the form of the Bayes factor for $H_1$ and $H_0$, respectively, when two-tailed hypotheses are used. When we look for intensity coding, we focus here on positive intensity coding, and thus use directed hypotheses, marked with $p_1$ or $BF_{+0}$ or $BF_{0+}$, with the + indicating a directed $H_1$, using conventions as in *Keysers et al., 2020*. It should be noted that the use of one-tailed statistics, which is sometimes criticized when exclusively using frequentist statistics, has important advantages when combining the frequentist with a Bayesian framework, in that it increases the sensitivity for falsifying the alternative hypothesis in a Bayesian framework. When multiple tests were performed across a high number (>1500) of adjacent frequency–time intersections or timepoints separately, we used cluster-based corrections that reveal large spectral and/or temporal windows of significance because the main focus of these analyses was to discover these critical windows, not the precise time- or frequency points. These cluster-based corrections are explained in detail in the section 'Intensity coding in LFPs.' On the other hand, when multiple tests were performed across a low number (<30) of adjacent timepoints, we used false discovery rate (FDR) corrections since these analyses aimed at finding the significance of precise timepoints, not that of wide temporal windows. When testing multiple neurons or multiple bipolar recordings, we do not correct for multiple comparisons when attributing a property to a location as this would result in changing the property of a location based on how many locations have been tested. Instead, we then examine whether the number of electrodes with a certain property exceeds the number expected by chance using binomial distributions. When performing Bayesian ANOVAs, we report $BF_{incl}$, which is the likelihood of models including a particular factor (or interaction of factors) divided by the likelihood of models excluding that particular factor (or interaction), as recommended by Rouder and coworkers and implemented in JASP (*Rouder et al., 2017*; *Rouder et al., 2016*; *Rouder et al., 2012*). When performing Bayesian *t*-tests in JASP, we use the default priors and methods proposed by *Rouder et al., 2009*. For nonparametric tests in JASP, we used the method described in *van Doorn et al., 2020*.

## Behavioral analyses

To explore whether patients were impaired in their ability to perform the task, our rationale was to consider the average rating of all control participants as the normative rating. We then compared the vector of 30 ratings (one per movie for 30 movies) of each member of the control group against the average of the other members of the control group to define a distribution of how far from the normative rating healthy volunteers tend to fall. For the patients, we compared their ratings against the average rating of the control group, and compared how similar patient ratings were to the normative average against the distribution of how similar left-out control participants are to the normative average. We calculated three metrics of similarity: the Spearman's rank-order correlation, the slope, and the intercept of a simple linear regressions between the ratings of each of the patients and the average rating of all control samples.

## Preprocessing of LFPs

To reduce artifacts and extract local signals, iEEG macrocontact recordings were digitally re-referenced in a bipolar layout (*Figure 1d*). This generated 85 bipolar recordings from 102 contacts in the insula, with patients having between 5 and 19 bipolar channels (*Figure 1c*, *Figure 1—source data 1*). Re-referencing attenuated 50 Hz noise sufficiently to omit digital filters that distort data. Continuous recordings were separated into trials of 4 s: 1 s premovie baseline, 2 s video, and 1 s postmovie. Trials were visually checked for ground failure and amplitude saturation (none was detected), downsampled to 400 Hz, and detrended.

## Time–frequency decomposition of LFPs

A sliding window Hanning taper-based approach was used for each trial with the following parameters: frequencies from 1 to 200 Hz in steps of 1 Hz; timepoints from –1 s (relative to movie onset) to 3 s in steps of 0.0025 s; and for each frequency, a single-sliding Hanning taper window with the duration of eight cycles (maximum = 1 s; minimum = 0.1 s). Trials were expressed as % power change relative to baseline (–1 s to 0 s) separately for each frequency: $y(t) = ((P(t) - P_0)/P_0)$, with $P_0$ = average of baseline. Points with $y(t) \pm 10$ standard deviations from the mean of the other trials were excluded to not reject entire trials, but only outlier time–frequency points in some trials (rejections were rare, mean rejected time–frequency points = 0.0032% ± 0.0035 SD).

## Intensity coding in LFPs

In the LFP signal, we consider that a bipolar channel shows intensity coding if its trial-by-trial power variations correlate positively with the variation in the pain intensity reported by the patient. We always coded the 1–2, 3–4, 5–6, and 7–8 rating options as 1, 2, 3, and 4. For each bipolar recording, we then calculated the Spearman's rank correlations (due to the ordinal nature of intensity ratings) between the patient's behavioral intensity ratings and the neural activity power estimate over all trials. In time–frequency-resolved analyses, these correlation analyses were conducted separately using the power estimates at each time–frequency intersection separately. In frequency band analyses, the correlation was calculated using the average of the power estimates within a specific frequency band. For both analysis types, a one-sample $t$-test was used to test whether the average correlation over the 85 bipolar recordings were greater than 0. Correlations were not Fisher $r$->$z$ transformed because $r$ and $z$ values remain virtually identical for $-0.5 < r < 0.5$, which is the range in which the correlations we examined remain. These time–frequency-resolved analyses were conducted separately for each of a large number of time–frequency intersections. Since we were interested in finding relatively large time–frequency windows of significance, these multiple tests were corrected using a modified version of the nonparametric cluster-based correction method described by *Maris and Oostenveld, 2007*. Specifically, we used a circular shift procedure that consisted of 1000 iterations to generate a null distribution. In each iteration, the time–frequency profile of the correlation coefficients for each contact was randomly shifted in the time domain. Then, all such time-shifted profiles were entered into an analysis, which, as described above, tested the significance of the correlation coefficients at each time–frequency intersection via a separate one-sample $t$-test against 0. Finally, the sum of the largest significant positive and negative clusters was calculated separately. At the end of the 1000 iterations, two separate null distributions were generated, one for positive and one for negative clusters, which showed the maximum sum of the significant clusters expected by chance. For statistical inference, the probabilities of the cluster sums observed in our data were calculated under the corresponding null distributions. The multiple tests in the time-resolved frequency band analyses were corrected based on exactly the same circular-shift and nonparametric cluster-correction method.

We performed a similar analysis to identify time–frequency bands with significant intensity coding when separating the Hand and Face trials. Note that including half the number of trials makes this analysis less powerful than the Hand and Face combined analysis, which is why we used the broad-band frequency band (20–190 Hz) resulting from the combined analysis. To directly compare the time–frequency profiles of intensity coding for Hand vs. Face trials, we performed a similar analysis for cluster correction, except that for each time–frequency point separately, the Hand and Face correlation coefficient distributions across the 85 electrodes were directly compared with a paired-samples $t$-test, and, instead of the circular shift randomization procedure, trials were randomly assigned as Hand and Face trials at each step of the 1000 iterations for generating the null distributions.

## Resampling LFPs from the entire brain

To test whether the BBP–rating association observed in the insular electrodes was enriched compared to what could have been observed in any 85 bipolar recordings anywhere in the brain, we made use of not just the insular but all the intracranial macroelectrodes that were implanted in the same patients (see *Figure 2—figure supplements 2 and 3* for anatomical locations of all macrocontacts). The recordings from these electrodes were preprocessed exactly as described for the insular electrodes. The seven patients included in these analyses had between 91 and 149 (mean ± SD = 114 ± 20) bipolar recordings distributed throughout the two hemispheres and various regions of the four brain lobes. The BBP–rating Spearman's correlation coefficients of each of these electrodes were entered into a resampling method, in which for each of the 100,000 iterations, from each patient, a random subset of these correlation coefficients were selected. The number that was selected for each patient was determined by the number of insular electrodes that patient had; that is, since patient A had five insular electrodes in the main analyses, five random electrodes were selected from the entire brain in these analyses. This was done to ensure that possible patient-specific biases in the analysis of insular electrodes were maintained in these analyses. This way, in each iteration, a total of 85 electrodes were selected randomly from the entire brain and tested with a one-sample $t$-test against 0. The resulting 100,000 $t$-values from all the iterations were used as the null distribution to test whether the $t$-value observed in the

insula was greater than what would be expected if we were not focusing on the insula and were randomly sampling from the entire brain. It is important to note that while the anatomical location of electrodes in and close to the insula was carefully determined, this manual procedure was not performed for electrodes clearly outside the insula, making it possible that some of these extrainsular electrodes included in this resampling were located in the white matter or cerebrospinal fluid, and this analysis should thus be considered with a grain of salt.

## Extracting shape and motion information from videos

A recent systematic review has revealed that facial expressions of pain are most consistently characterized by lowering of the brow and tightening of the eyelid (**Kunz et al., 2019**), corresponding to facial AUs 4 and 7 (**Ekman and Friesen, 1978**; inlet in **Figure 3a**). More specifically, research has evidenced that people fall into four clusters that differ in how they express pain (**Kunz and Lautenbacher, 2014**), and our protagonist fell within cluster IV, who express pain by furrowing brows and tightening eyes, but not opening the mouth or wrinkling the nose. Accordingly, we quantify the painfulness expressed in the shape of our protagonist's face based on facial AUs 4 and 7. To get a replicable and objective measurement of these AU, we used the FaceReader software (Noldus, the Netherlands), which uses a deep convolutional neural net to automatically extract the level of activation of the classic facial action units (**Ekman and Friesen, 1978**). FaceReader reliably obtained estimates for the facial AUs 4 and 7 from all but three frames from our 30 movies, and we thus quantified the pain-related shape information contained in each frame of our movies as the average of AUs 4 and 7. When applied to the frames used in our psychophysical experiment mentioned above, the average activation of AUs 4 and 7 correlated at $r_p = 0.95$ with the average rating from human observers of the same static images, validating the utility of this automated signal. Unfortunately, we found no software that could estimate muscular contraction from the hand in a similar way, and we thus did not see an obvious way to extract shape information from the Hand stimuli. Given that participants are also very poor in their ability to rate painfulness from static frames of the hand configuration in our stimuli, as shown by our psychophysics, we felt that not quantifying shape information for the Hand stimuli was acceptable.

To quantify motion over time for each video, we use motion energy, an established and objective way to extract dynamics from any movie, using the average of the Euclidean distances between the RGB vectors of the corresponding pixels across every two consecutive frames.

## PLSR decoding of intensity coding from shape and motion information

To identify when motion or shape information may contribute to predicting the overall intensity rating R of the movie i, we used PLSR analyses using the 'plsregress' function in MATLAB, with a single component. For the Hand, where no shape information was available, the predictor for each movie i was the motion M at frame t, and the plsregress thus identified the weights (B), such that $R(i) = M(i,t) B(t) + B_0(i)$. For the Face, where both motion and shape information S was available, as the average of AUs 4 and 7 in each frame, we concatenated M(i,t) and S(i,t) into a single predictor X to identify weights such that $R(i) = XB + B_0(i)$. We used PLSR here in particular because both M and S have high temporal autocorrelations and are mutually correlated, and PLSR is well suited for such cases.

Second, we can use the PLSR method to see how accurately the motion and/or shape profile across the entire movie can be used to predict the rating of the patients using a cross-validation. The predictive accuracy for both the motion and shape time course was calculated in 1000 iterations. In each iteration, the videos were randomly divided into three equal-sized samples. For each of these three samples separately, the remaining two samples were used to calculate PLSR beta coefficients, which were then used to predict the ratings of our patients. The Pearson's correlation coefficient between the actual ratings and the predicted ratings was taken as measures of predictive accuracy and averaged across all iterations. A similar procedure was also applied to calculate the null distribution of such correlation coefficients, in which the 1000-iteration step described above was repeated for 10,000 times, each time with a different randomly shuffled version of the observed ratings. The probability of the observed decoding accuracies was then estimated by ranking the accuracy based on the actual ratings within the distribution of shuffled ratings. **Figure 3d–f** shows that motion and shape information each allows one to predict movie ratings with high accuracy.

## Probability of Face-but-not-Hand, Hand-but-not-Face, and dual-intensity coding LFPs

Correlations between BBP and rating were thresholded as significant or as providing evidence of absence as follows (*Figure 4b*). At n = 60 trials, values above $r = 0.214$ show a significant positive association ($p_1 < 0.05$). Values below $r = 0.085$ provide evidence for the absence of a positive association ($BF_{+0} < 1/3$). Intermediate values are inconclusive (*Keysers et al., 2020*). Both the frequentist and Bayesian criteria we use here are subject to type I/II errors, and we thus asked whether the number of bipolar recordings we find in these quadrants is above what we would expect by the probability of these errors. For the frequentist criterion, $p_1 < 0.05$, we expect 5% of locations to be classified as showing significant intensity coding even if $H_0$ was true (i.e., despite no real intensity coding). With regard to the dual-coding quadrant that we are interested in, two types of errors could be made. The most likely misclassification is for a location showing one intensity coding to be mistakenly classified as having dual intensity coding. To test whether we have above-chance numbers of dual-coding locations, we thus take all the locations with Hand intensity coding (21/85), and ask among those whether finding 5 also showing Face intensity coding is more than what we expect using a binomial (n = 21, $k_{success} = 5$, $\alpha = 0.05$) and the results showed clear evidence that there are more locations also representing the Face among the Hand locations ($p_1 = 0.003$, $BF_{+0} = 17.09$). The same could be done by looking whether 5 Hand intensity coding is overrepresented among 15 Face intensity coding locations ($p_1 = 0.0006$, $BF_{+0} = 117$). A less likely misclassification is for a location that shows neither intensity coding to be classified as having both ($\alpha = 0.05^2$). Making five such misclassifications among 85 recordings is also highly unlikely ($p_1 = 3 \times 10^{-6}$, $BF_{+0} = 4446$). For the Bayesian criterion, $BF_{+0} < 1/3$ this calculation is more difficult to perform because Bayesian criteria are not defined directly based on a false-positive rate. However, *Jeffreys, 1939* chose the bound of $BF < 1/3$ as evidence of absence or presence precisely because it roughly corresponds to a $p = 0.05$ with a standard prior on the effect sizes in $H_1$. We can thus, as a reasonable approximation, assume that if $H_1$ is actually true, and a location thus shows significant positive intensity coding, only 5% would be falsely classified as showing evidence against $H_1$. With that approximation, among the 21 Hand intensity coding locations, it is highly unlikely to encounter 10/21 showing evidence that they do not encode the Face if in reality they did: (binomial with n = 21, k = 10, $\alpha = 0.05$, $p_1 = 2 \times 10^{-8}$, $BF_{+0} = 2 \times 10^6$). Even if the false rejection rate were much higher (e.g., $\alpha = 0.25$), 10/21 remain unlikely (binomial $p_1 = 0.025$, $BF_{+0} = 4.2$). Similarly, among the 15 locations with significant Face intensity coding, finding 6 with evidence for not encoding the Hand is again unlikely (binomial n = 15, k = 6, $\alpha = 0.05$, $p_1 = 5 \times 10^{-5}$, $BF_{+0} = 1334$). We can thus conclude that preference is over-represented in the insula compared to what we would expect if all neurons showing a preference for one stimulus type would also show coding for the other.

The same analysis was applied to an exemplar participant (*Figure 4d*). Using the same logic in that patient, we find that the number of Hand-but-not-Face coding locations ($p_1 = 2 \times 10^{-4}$, $BF_{+0} = 701$) and the number of Face-but-not-Hand coding locations ($p_1 = 0.007$, $BF_{+0} = 37$) are surprising, but the number of dual-coding locations (1/15) is not surprising among the three Face ($p_1 = 0.143$, $BF_{+0} = 1.9$) or seven Hand ($p_1 = 0.3$, $BF_{+0} = 0.48$) coding locations.

## PLSR decoding of intensity from the LFP insula activity pattern

To explore how well the pattern of activity across all 85 bipolar recordings reflects the perceived intensity reported by our patients, we applied a PLSR regression approach similar to that used to infer how well shape or motion predicts ratings, except that instead of using motion over 50 frames, we used BBP over 85 electrodes. Specifically, BBP across the 85 sites, averaged over the early period for Hand videos and in the late period for Face videos, were separately used as predictors in two separate PLSR analyses predicting the participants' average pain ratings. The decoding accuracies were each calculated in 1000 iterations. In each iteration, the videos were randomly divided into three equal-sized samples. For each one of these three samples, the remaining two samples were used as a training set to calculate PLSR beta coefficients, which were then used to predict the ratings in the remaining test sample. The Pearson's correlation coefficient between the predicted ratings and the actual ratings of the patients was then taken as a measure of decoding accuracy and averaged across all iterations. We used Pearson here because the data was normally distributed, and we compared two ratings. A similar procedure was also applied to calculate the null distribution of such correlation coefficients, in which the 1000-iteration step described above was repeated for 10,000 times, each time with a different

randomly shuffled version of the original ratings. The probability of the observed decoding accuracies was estimated under the corresponding null distributions as the rank of the actual average accuracy against the shuffled accuracies. We first performed this analysis within each stimulus type; that is, we trained and tested on Hand stimuli or we trained and tested on Face stimuli. Then, to explore whether the pattern of activity could generalize across stimulus type, we also performed cross-decoding analyses where we trained on one stimulus type (e.g., we determined the PLSR weights using ⅔ of Hand stimuli) and then tested them on the other (e.g., predicted 1/3 of the Face stimuli). We first performed this analysis using a single PLSR component and found a trend (Hand: $r_{P(8)} = 0.281$, $p_1=0.093$; Face: $r_{P(8)} = 0.3$, $p_1=0.071$). Using two components in the PLSR analyses improved results, which now became significant for the Hand ($r_{P(8)} = 0.575$, $p_1=9 \times 10^{-4}$) and near significant for the Face ($r_{P(8)} = 0.331$, $p_1=0.058$). Increasing to three or four components did not further improve this level of decoding. We thus report the results using two components (*Figure 4e*) and used two components also for the cross-stimulus decoding, which turned out significant in both directions.

## Resting state connectivity analysis

To interrogate what connectivity profile is characteristic for electrode pairs with high-intensity coding, we used Neurosynth (*Yarkoni et al., 2011*) to extract a whole-brain resting state connectivity map for the MNI location of each of the 85 contact pairs in the insula (*Figure 5—source data 1*). Using SPM12 (https://www.fil.ion.ucl.ac.uk/spm/software/spm12/), we performed a regression analysis (general linear model) that included the 85 voxel-wise resting state connectivity maps and two predictors: the correlation between power and rating for the Hand in the early window, and for the Face in the late window. Results were thresholded at p<0.001, corrected for multiple comparisons using family-wise error correction at the cluster level. Results were then illustrated on an inflated cortical template provided in SPM12, and significant voxels were attributed to specific brain regions using the Anatomy toolbox 3.0.

## Spike sorting and selection of responsive single units

Three patients had microwires (Behnke–Fried electrodes, Ad-Tech Medical; *Fried et al., 1999*) in the insula protruding from the electrode tip (plusses in *Figure 1c*). Spikes were detected and sorted using Wave_Clus2 (*Quiroga et al., 2004*). In short, raw data was filtered between 300 and 3000 Hz. As per default settings, spike waveforms were extracted from 0.625 ms before to 1.375 ms after the signal exceeded a 5*noise threshold, where noise was the unbiased estimate of the median absolute deviation. Wave_Clus2 sorted and clustered the waveforms automatically and were manually checked by author RB. Clusters were excluded in which >2% of spikes were observed with an interspike interval < 2 ms or with firing rate < 1 Hz. To identify cells that responded to our stimuli, we used a Wilcoxon signed-rank test comparing spike counts during baseline (–1 s to 0 s) against that during the pain period (1–2 s) for Hand and Face trials together. Only cells that showed a response to the stimuli ($p_1<0.05$), irrespective of pain intensity, were considered for further analysis.

Similar to LFP analyses, a cell was said to show intensity coding if spike counts rank correlated positively with reported intensity. Because JASP includes Bayesian statistics using Kendall's tau but not Spearman's *r*, we used the former to quantify evidence for or against intensity coding.

## Broadband power analysis in microelectrodes

To explore whether intensity coding in cells and the BBP (20–190 Hz; *Figure 2a*) from the same microwire were related, for the 10 microwires that yielded responsive neurons (whether these neurons showed intensity coding or not) we quantified the association between BBP averaged over the pain period (1–2 s) and intensity ratings (1–2, 3–4, 5–6, 7–8) using rank correlation coefficients separately for Face and Hand videos (again using Kendall's tau to provide $BF_{+0}$ estimates). All eight microwires protruding from the same electrode were first re-referenced to the microwire with the least spiking and lowest artifacts, yielding seven microwire recordings for each of the four electrode tips with wires in the insula. Data were filtered to remove 50 Hz noise and harmonics at 100 and 150 Hz. Subsequently, they were separated into trials of 4 s (–1 s to 3 s relative to video onset), downsampled to 400 Hz, and visually checked for artifacts. The time–frequency decomposition of power followed the same procedure as for the macrocontact recordings. Finally, intensity coding at the level of spikes

(i.e., $r_K$(spikes,rating)) and BBP ($r_K$(BBP,rating)) from the same wire was compared using a Kendall's tau coefficient.

## fMRI experiment

### Participants

Twenty-five healthy volunteers participated in the study. The full dataset of two participants was excluded from the analyses because head motions were above 3 mm. Analyses were performed on the remaining 23 participants (13 females; mean age = 28.76 years old ± 6.16 SD). The study was approved by the local ethics committee of the University of Amsterdam (project number: 2017-EXT-8542).

### Stimuli and procedure

The video pain rating task was performed as described in the section 'Video rating task,' but with the following differences. Each trial started with a gray fixation cross lasting 7–10 s, followed by a red fixation cross for 1 s, followed by the 2 s video, followed by a red fixation cross lasting 2–8 s, followed by the rating scale ranging from 'not painful at all' ('0') to 'most intense imaginable pain' ('10'). The design also includes another condition we will not analyze here, in which participants viewed videos varying in color saturation and had to report on a scale from 'not a change' ('0') to 'a very big change' ('10'). Participants were asked to provide a rating by moving the bar along the scale using two buttons for right and left (index and middle finger) and a third one for confirming their response (ring finger) using their left hand. The direction of the scale and the initial position of the bar was randomized in each trial. The videos used for the Face and Hand conditions for the electrophysiology and fMRI experiment were generated in the same way but were not identical. The task was split up into six blocks of 30 trials each: two blocks of electrical pain stimulations, two blocks of mechanical slaps by a belt, and two blocks of videos with changes in color saturation (presented in 46 separate fMRI acquisition runs). Anatomical images were recorded between the fourth and fifth run of fMRI acquisition.

### Data acquisition

MRI images were acquired with a 3-Tesla Philips Ingenia CX system using a 32-channel head coil. One T1-weighted structural image (matrix = 240 × 222; 170 slices; voxel size = 1 × 1 × 1 mm) was collected per participant together with echoplanar imaging (EPI) volumes (matrix M × P: 80 × 78; 32 transversal slices acquired in ascending order; TR = 1.7 s; TE = 27.6 ms; flip angle: 72.90°; voxel size = 3 × 3 × 3 mm, including a 0.349 mm slice gap).

### Data analysis

#### Preprocessing

MRI data were processed in SPM12. EPI images were slice-time-corrected to the middle slice and realigned to the mean EPI. High-quality T1 images were co-registered to the mean EPI image and segmented. The normalization parameters computed during the segmentation were used to normalize the gray matter segment (1 × 1 × 1 mm) and the EPI images (2 × 2 × 2 mm) to the MNI templates. Finally, EPIs images were smoothed with a 6 mm kernel.

#### Voxel-wise analysis

At the first level, we defined separate regressors for the Hand videos, Face videos, rating scale, and button presses. Analyses focused on the 2 s that the videos were presented. The trial-by-trial ratings given by the participants were used as a parametric modulator, one modulator for the Face and one for the Hand trials, on the respective video regressor. The rating-scale regressor started from the moment the scale appeared and ended with participants' confirmation button press. The button-press regressor, finally, had zero duration events aligned to the moment of each button press. Six additional regressors of no interest were included to model head movements. To quantify the degree to which each voxel in the insula had BOLD activity associated with trial-by-trial ratings, we then brought the parameter estimate for the parametric modulator obtained for Hand and Face trials separately to a second-level *t*-test with 23 participants, and then used the resulting *t*-value as a measure of the random effect size of the association for each voxel. We used *t*-values rather than the average value of the parametric modulator because these values are to be compared against out-of-sample values

of patients, and the topography of *t*-values is a better predictor for out-of-sample generalizations. However, the average parameter value correlated above 0.9 with the *t*-value.

## Multivariate regression

To investigate whether the pattern of BOLD activity across all voxels in the insula encodes intensity, we additionally performed a multivariate regression analysis akin to the PLSR for the BBP described in above. For each participant, we performed a general linear model that estimated a separate parameter estimate for the video epoch of trials in which participants gave a rating of 0–2, 3–4, 5–6, or 7–8, respectively, separately for Hand and Face trials. In MATLAB, we then loaded for each subject the parameter estimate images for each level of rating and only included voxels that fell within our insula mask (*Faillenot et al., 2017*). We then trained a weighted PLSR using the MATLAB function plsregress and the data from all but one participant to predict rating based on a linear combination of the parameter estimates in each voxel and used this linear combination to predict the rating of the left-out participant, repeating the procedure for each participant. We weighted the regression by replicating each parameter estimate image in the training and testing set by the number of trials that went into it. We quantified how accurately the regression predicted the rating of the left-out participants using Kendall's tau, then tested whether the performance was above chance by comparing the 23 prediction accuracies (Kendall's tau) against zero in a one-tailed test. Based on the analysis on the BBP, we performed this analysis with procedure using two or three components.

## Acknowledgements

We thank Pieter Roelfsema for enabling the collaboration that led to the access to the patients, Eline Ramaaker for her assistance in electrode localization, Agneta Fischer and George Bulte at UvA for advice and help with the use of FaceReader, and Tess den Uyl for advice on how to use FaceReader specifically to analyze facial expressions of pain. This work was supported by Dutch Research Council (NWO) VIDI grant (452-14-015) to VG and VICI grant (453-15-009) to CK.

## Additional information

### Funding

| Funder | Grant reference number | Author |
|---|---|---|
| Nederlandse Organisatie voor Wetenschappelijk Onderzoek | 452-14-015 | Valeria Gazzola |
| Nederlandse Organisatie voor Wetenschappelijk Onderzoek | 453-15-009 | Christian Keysers |

The funders had no role in study design, data collection and interpretation, or the decision to submit the work for publication.

### Author contributions

Efe Soyman, Data curation, Software, Formal analysis, Validation, Visualization, Writing – original draft, Project administration, Writing – review and editing; Rune Bruls, Kalliopi Ioumpa, Data curation, Software, Formal analysis, Validation, Investigation, Visualization, Writing – original draft, Project administration, Writing – review and editing; Laura Müller-Pinzler, Software, Formal analysis, Validation, Writing – original draft, Writing – review and editing; Selene Gallo, Resources, Software, Formal analysis, Validation, Writing – review and editing; Chaoyi Qin, Visualization, Writing – review and editing; Elisabeth CW van Straaten, Matthew W Self, Judith C Peters, Jessy K Possel, Investigation, Writing – review and editing; Yoshiyuki Onuki, Writing – review and editing; Johannes C Baayen, Sander Idema, Resources, Investigation, Writing – review and editing; Christian Keysers, Valeria Gazzola, Conceptualization, Resources, Software, Formal analysis, Supervision, Funding acquisition, Visualization, Methodology, Writing – original draft, Writing – review and editing

## Author ORCIDs
Efe Soyman  http://orcid.org/0000-0003-0192-1541
Laura Müller-Pinzler  http://orcid.org/0000-0002-5567-5430
Matthew W Self  http://orcid.org/0000-0001-5731-579X
Christian Keysers  http://orcid.org/0000-0002-2845-5467
Valeria Gazzola  http://orcid.org/0000-0003-0324-0619

## Ethics

Human subjects: Written informed consent was obtained from each participant before participating in the study. All procedures on patients were approved by the medical ethical committee of the Vrije University Medical Center (protocol 2016/037). All procedures on healthy participants were approved by the local ethics committee of the University of Amsterdam (protocols 2017-EXT-8542 and 2021-EXT-13608). In addition, written informed consent to publish was obtained from the individual whose photographs are shown in Figures 1 and 3 of the article.

## Decision letter and Author response

Decision letter https://doi.org/10.7554/eLife.75197.sa1
Author response https://doi.org/10.7554/eLife.75197.sa2

---

# Additional files

## Supplementary files
• Transparent reporting form

## Data availability

The data presented in this work is publicly available at the Open Science Framework: https://osf.io/mcahz/.

The following dataset was generated:

| Author(s) | Year | Dataset title | Dataset URL | Database and Identifier |
|---|---|---|---|---|
| Gazzola V | 2021 | Intracranial Human Recordings Reveal Intensity Coding for the Pain of Others in the Insula | https://osf.io/mcahz/ | Open Science Framework, mcahz |

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

# Appendix 1

## How shape information influences participant's rating

We collected data from a sample of 40 healthy participants in an online frame rating task to assess whether participants can recognize pain intensity from static frames taken at the key moment of the Face and Hand videos. *Appendix 1—figure 1* shows that the rating of single frames of Faces was even slightly more consistent than the ratings of the entire videos from which they were taken (i.e., $r_S(frame_i,AV) > r_S(movie_i,AV)$, $W = 193$, $p_2=0.003$, $BF_{10} = 31.29$). In contrast, for Hands, the rating of the frames was poor compared to the rating of the movies (i.e., $r_S(frame_i,AV) < r_S(movie_i,AV)$, $t_{(38)} = 11.96$, $p_2=2 \times 10^{-14}$, $BF_{10} = 4 \times 10^{11}$). Directly comparing the change of performance across the two stimulus types as an interaction of an effector (Hand vs. Face) × stimulus (Movie vs. Frame) ANOVA revealed a highly significant effect ($F_{(1,38)} = 178.98$, $p=6 \times 10^{-16}$, $BF_{incl} = \infty$). Finally, because for Hands, the accuracy was low, we also tested whether the accuracy was above zero, and it was ($W = 580$, $p_2=0.022$, $BF_{10} = 6.19$). Hence, for Faces, static shape information was sufficient to explain the rating of the videos, while for Hands, the shape information in the frames we selected was not sufficient. It should be noted that in principle information contained in other frames may have contained useful information, but informal reports of some participants confirmed that they paid attention more to kinematic than configurational cues.

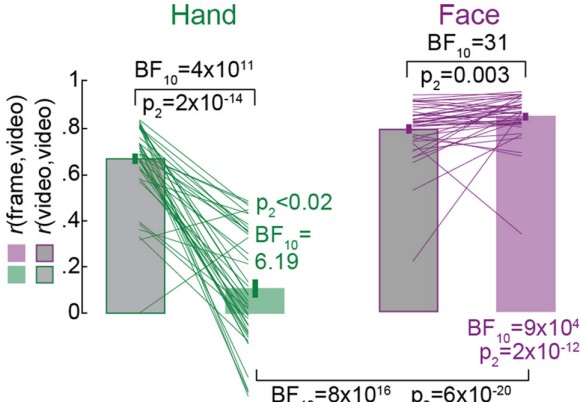

**Appendix 1—figure 1.** Rating from shape information alone. Mean ± SEM correlation coefficients between each participant's ratings in the online *frame* rating task and the average ratings of the other participants in the online *video* rating task ($r_S(frame,average\_video)$, green and purple) compared against that between participant's ratings in the online *video* rating task and the average ratings of the other participants in the same task ($r_S(video,average\_video)$, gray) for Hands and Faces separately. Black statistics above the bars compare the respective frame and video ratings, the colored statistics compare the frame ratings against zero. The black statistics under the bars compare the frame ratings between Hands and Faces.

