## [Editor Report]

This fundamental work shows that insular broadband activity (20-190 Hz) correlates with the perceived intensity of others' painful experiences viewed as movies. This finding is not only important to the pain field but also constitutes an important contribution to the neuroscience of empathy. However, whereas the intracranial recording data are compelling, a limitation of the study remains the lack of specific control stimuli to determine whether the early broad-band insular response when viewing the hand could be related to the observation of body movement rather than "intensity coding for the pain of others", and whether the late broadband response to a facial expression of pain might also be observed for other facial expressions.

---

## [Decision Letter]

**Decision letter after peer review:**

Thank you for submitting your article "Intracranial Human Recordings Reveal Intensity Coding for the Pain of Others in the Insula" for consideration by *eLife*. Your article has been reviewed by 3 Reviewers, and the evaluation has been overseen by Drs. Shackman (Reviewing Editor) and Büchel (Senior Editor).

The Reviewers and Dr. Shackman discussed the critiques, and he has drafted this to help you prepare a revised submission.

Essential revisions:

The 3 Reviewers expressed some enthusiasm for the report, noting that

– The paper addresses an important research question which is investigated in considerable detail.

– The authors are to be commended for the care and work they put into this manuscript.

– The study has much to recommend it. The iEEG data are rare and difficult to acquire, and the authors performed a number of interesting and creative analyses, including analyses of correlations with subjective rating, timing of pain rating-correlated iEEG signal relative to facial and motion information in the movies (and assessments of independent raters), and analyses of functional connectivity leveraging Neurosynth and a previously unpublished fMRI study.

– This study provides a very thorough analysis of insular activity recorded from 7 human participants while they viewed depictions of others in pain, evidenced by movies of both facial expressions made by a model or a hand being hit by a belt. These results provide a rare window into the neural underpinnings of the perception of others' pain, which is a critical ingredient of empathy and human sociality.

– The manuscript reports a highly original study using intracerebral recordings performed in humans for the diagnostic workup of epilepsy to explore whether and how the insula may encode pain experienced by others. The obtained results, showing that the magnitude of insular broadband activity (20-190 Hz) correlates with the perceived intensity of painful experiences viewed as movies (either facial expressions of pain or painful stimuli applied onto the hand), constitute an important contribution to the neuroscience of empathy.

– A strength of the manuscript is that, to better understand the time course of the elicited responses and their relation to perceived intensity and take full advantage of the temporal resolution of iEEG recordings, the authors conducted an in-depth analysis of the motion and shape information contained in each video clip, to understand how variations in these stimulus features as a function of time could be used by the participants to decode intensity of the viewed experience. Most importantly, the authors also characterized the relationship between the temporal dynamics of these stimulus features (motion and shape) and the time course of the correlation between insular activity and pain ratings.

– Assuming the "trend-level" responses related to pain facial expressions are reliable, there are several other interesting characteristics that emerged from the analyses. The analyses suggested overlapping, but separable, distributions of insular locations that encode pain from hands, faces, or both. This is consistent with work on population coding in other areas, and suggests (as the authors argue) that signals at many locations cannot be reduced to "salience" in general as they code for pain inferred from specific stimulus types. These results add to the literature, and appear to correspond with other fMRI studies that have examined intensity-coding of perceived pain. For example, Krishnan et al. 2016, *eLife* found that among individual brain areas that predict intensity of perceived pain from pictures of hands and feet, the insula was among the most strongly predictive. (They also found that a distributed network including other brain regions as well was much more strongly predictive). Zhou et al. 2020 *eLife* studied perceived pain from both facial expressions and pictures of body parts. They identified an overlapping area of the mid– and anterior insula that predicted perceived pain across both stimulus types. That area may be similar to the locations with overlapping encoding observed here, and the distribution across the insula of differentially predictive signals for body parts and faces may be similar to the distribution observed here. Both these studies analyzed relationships between brain activity and trial-by-trial ratings of perceived pain, and so are directly comparable.

– The present results are also consistent with earlier studies that did not test the relationship with perceived pain, but tested multiple types of stimuli related to pain or other emotions. Corradi-Dell'Acqua et al. 2011 studied fMRI responses to pain-related, non-painful but threatening images, and neutral images, and found responses in the insula to both types of negative images. Later, Corradi-Dell'Acqua et al. 2016 extended this overlap analysis to local multivariate patterns of activity in response to shock pain and disgusting tastes administered to self and other, and to perceived unfairness in the Ultimatum Game. They found evidence for common patterns, particularly in the right anterior insula. Based on these studies, it would be interesting to see whether the iEEG signals recorded in this study would respond similarly to other types of aversive stimuli, including somatic pain. That would inform the field on whether they are related to pain perception or to another, correlated affective state. Though the authors rightly argue that the differential encoding of perceived pain implies that the entire insula cannot simply be encoding "salience". However, neurons respond to complex configurations of properties, and it may be possible to find signals in the insula or elsewhere that respond to many different combinations of stimulus properties, including conditional ones (e.g., only aversive stimuli delivered to the hand) without truly "representing" or encoding the perception of pain per se. Conclusively identifying a representation of perceived pain, or any other construct, is a noble but difficult challenge, however, and this work takes a step in this direction.

Narrative Clarity

– The manuscript is very dense, making it hard to follow. Some sections with many

statistics could be organized into tables. It was sometimes difficult to follow what was done, particularly in the fMRI section. The main paper has several analyses that really seem to be supplementary and are somewhat distracting (e.g., a long section in the main text on ratings of external observers of the stimuli making ratings).

Scholarship/Past Work

– In several places, particularly in the introduction, a small set of overlapping papers are cited that draw heavily on the authors' prior work, but omit some of the most critical other papers – Krishnan et al. 2016 is not cited at all, and Zhou et al. 2020 is mentioned only in passing, though it is the most similar paper in the literature in design and scope. Corradi dell'Acqua's papers are not cited. Most of the fMRI studies cited in the introduction look at the relatively coarse contrast of observed pain versus control, but do not identify signals that encode the intensity of pain within-person. The studies that have done this most directly are Krishnan et al. 2016 and Zhou et al. 2021. Both identify areas within the mid-insula that encode perceived pain.

Study Framing

– The paper is framed in the introduction around comparisons with direct pain experience, but there is no comparison with pain experience here, so this is somewhat misleading. The most relevant prior papers are not discussed.

Approach

– Key details about the non-insula recordings are omitted

A nice feature of the manuscript is the comparison of insular electrodes random electrodes "throughout the brain", but what electrode locations were available, in how many individuals, and what is their distribution? Surely there are not electrode placements everywhere in the brain. Please clarify.

Approach/Results

– fMRI study is hard to follow and key details are omitted

One reviewer noted: Section 2.7 is very difficult to follow. I would very much welcome a careful rewrite of this section.

Another reviewer wrote that: The concept of the analyses using Neurosynth and fMRI are clever, but the fMRI study in particular is confusing. From the main text it is unclear how many conditions there were in the fMRI study (fMRI during both faces and hands? With ratings?), the sample size, what analyses were done ("leave one subject out" is mentioned but not explained), and whether fMRI activity predicts intensity for either faces or hands. It is stated that "performance did not differ across Hands and Faces", but not that performance was significant for either or, indeed, what analysis was done.

Approach/Results/Discussion

– Unit recordings. Data from 28 unit recordings were provided as additional evidence of pain-of-other coding. This is a very low number of units across only 3 patients, especially given that the results are based on 13 units that showed higher firing during the pain period compared to the baseline period. These observations should probably be relegated to the supplement. At minimum, the wording related to these findings should be revised to reflect the exploratory nature of these results. For example, I do not believe these results warrant labels such as "hand/face specific".

– Faces vs. hands effects. There do not seem to be significant brain correlations within faces alone that survive correction for multiple comparisons. Figure 3 shows a "trend"-level result, not significant. Could this indicate a "hand vs. face" effect that appears as a correlation in Figure 2? If hands are rated higher than faces, and hands produce greater BBP in the insula, then any electrode that responds to hands more than faces will show up as a correlation between brain and rated intensity. The way to test this would be to test and show correlations within hands and faces separately, but these were apparently not significant for faces. At minimum, the authors need to frankly address this concern in the Discussion

– Faces vs. hands effects . A main point of Figure 3 is that correlations with intensity ratings occur later for faces than hands. But the face signals are not really significant, which makes this kind of conclusion difficult, and there don't seem to be any inferential statistics on the timing of the effect. So it's unclear the timing difference is a real effect. Even though this is "purely explorative" it seems hard to justify an entire main figure and section of the text.

– Faces, hands, and bayes. The facial results appear to hinge on "trend-level" results in several places, and in part on an analysis of lack of differences between faces and hands. It's less desirable to make inferences about the lack of an interaction (a null effect; Figure 3). This is used to argue that there's "no difference between hand and face in the late period". Another interpretation is that there's simply no correlation with intensity for faces. There is some evidence for an effect, but the study perhaps doesn't have enough power to tell for sure. It is helpful that Bayes Factors are reported, but what threshold do the authors think is required to demonstrate evidence for the null in the presence of these multiple tests? (And what is BFincl as opposed to BF10?)

Discussion

– Frank and complete discussion of latencies. The authors report timing effects in the range of 40-320 ms. They cite the well-known study by Krolak-Salmon and colleagues. But it is difficult to know what to make of the current results. On the one hand, latencies on the range of 200-300 ms (across the brain) would be compatible with some studies (but not others). How does one interpret latencies as low as 40-60 ms in the insula, which presumably are not really feasible? On the other hand, the results in the 560-1000 ms range are consistent with many studies but are discussed as "too slow". Authors need to discuss the likely source of these latencies.

Discussion/Abstract

– Experimental approach does not warrant strong claims of specificity.

A Reviewer noted that: A question that one could raise – especially for the hand condition – is whether part of the trial-by-trial correlation between insular activity and ratings was the consequence of a relationship between that activity and low-level motion/shape features of the video clips, rather than a reflection of the insula being involved in "intensity coding for the pain of others". Fortunately, this question was at least partly addresses with the available data, by showing that the recorded insular activity correlated with motion energy when the motion was associated with pain (during the slapping of the belt), but not while the motion was innocuous (during the initial lifting of the belt). Nevertheless, as mentioned by the authors themselves, future experiments using additional control stimuli such as stimuli matched in terms of motion content but differing in the emotions they convey are critical to better understand the specificity of the described insular responses.

S/he also noted that: The observation that insular activity correlated with motion energy when motion was associated with pain, and not during the initial lifting of the belt is indeed an indication that the insular responses cannot be reduced to motion detection. However, in these video clips, the first second showed the belt lifting without any movements of the hand. Then, when the belt came down to hit the hand, this was followed by a reaction/movement of the hand. Therefore, one may wonder whether part of the recorded activity could be related to the specific observation of body movement. This might also relate to the observation that stronger insular activity was observed in the hand condition as compared to the face condition might also bodily movements

A different Reviewer noted that: The study makes use of a fantastic stimulus set from the research group. But if I understand the set correctly, the same actor is used for different levels of pain (face– and hand-based). While this is an excellent set to study some aspects of coding related to stimulus "preference" I don't believe it's general enough to establish "selectivity". Accordingly, I think it would be more reasonable to avoid the claims of "selectivity".

A third Reviewer wrote that: Notably, the authors do not intend to interpret the generalizability to direct pain experience or other affective states, or specificity to pain compared with other affective conditions; but without more information about these, it is difficult to tell what the signals in the insula actually represent. It leaves open the possibility that there is another, better explanation for activation of insular neurons than perceived pain. e.g., It could be about representation of the body more generally, rather than perceived pain specifically. Could they be coding for choices (ratings) themselves? Baliki found that insular activity correlates with rating intensity of a simple visual stimulus, and other studies (e.g., Grinband et al., Neuron) have found that the insula correlates with simple perceptual magnitude decisions. This will be an ongoing project for future work.

S/he noted that: The stimulus set chosen was limited, and appears to relate to one specific type of painful stimulus on one hand model, and one set of facial expressions made by one female model. As it's well known in general that neurons often have complex receptive fields, it's unclear whether other types of painful hand stimuli or facial expressions made by other models would yield similar findings. Perhaps the insula would respond more strongly to other faces – or perhaps not at all. Other studies have shown that women's pain is discounted (Zhang et al. 2021). The need for diverse sets of stimuli to establish relationships with brain activity that are not stimulus-specific is becoming increasingly recognized (e.g., Yarkoni 2020, Westfall and Yarkoni).

---

## [Author Response]

Essential revisions:Narrative Clarity– The manuscript is very dense, making it hard to follow. Some sections with manystatistics could be organized into tables. It was sometimes difficult to follow what was done, particularly in the fMRI section. The main paper has several analyses that really seem to be supplementary and are somewhat distracting (e.g., a long section in the main text on ratings of external observers of the stimuli making ratings).

Following the reviewer’s suggestion, we now moved some of the reported statistical values in tables, rather than in the main text. In particular, we added Table 2 and Table 3. We then introduced more information about the fMRI study already in the main text and moved the rating data obtained when observing shape information alone in Appendix 1 and Appendix 1 – figure 1. For the fMRI section, we now write:

“Finally, we leveraged existing unpublished fMRI data from our lab to in healthy participants to test whether the spatial gradient observed in the iEEG data resembles the spatial distribution of voxels in the insula correlating with intensity ratings. Twenty-three independent healthy volunteers participated in the fMRI-adapted version of the rating task. As for the iEEG experiment, participants were asked to rate Hand and Face 2 s videos on painfulness. The stimuli depicted the same actor as in the iEEG experiment and were made following the same procedure. To test whether the pattern of BOLD activity in the insula can be used to predict the ratings of the participants, we defined eight separate regressors in each participant, capturing all trials that participants rated with intensity 0-2, 3-4, 5-6 or 7-8, separately for Hand and Face trials (eight in total). […]”.

For the appendix, we now write:

“Appendix 1

How shape information influences participant’s rating

We collected data from a healthy sample of 40 healthy participants in an online frame rating task to assess whether participants can recognize pain intensity from static frames taken at the key moment of the Face and Hand videos. Appendix 1 – figure 1 shows that the rating of single frames of Faces were even slightly more consistent than the ratings of the entire videos from which they were taken (i.e., *r_S_*(frame_i_,AV)>*r_S_*(movie_i_,AV), *W*=193, *p_2_*=0.003, BF_10_=31.29). In contrast, for Hands, the rating of the frames was poor compared to the rating of the movies (i.e., *r_S_*(frame_i_,AV)<*r_S_*(movie_i_,AV), *t_(38)_*=11.96, *p_2_*=2x10^-14^, BF_10_=4x10^11^). Directly comparing the change of performance across the two stimulus types as an interaction in a effector (Hand vs Face) x stimulus (Movie vs Frame) ANOVA revealed a highly significant effect (*F_(1,38)_*=178.98, *p*=6x10^-16^, BF_incl_=∞). Finally, because for Hands, the accuracy was low, we also tested if the accuracy was above zero, and it was (*W*=580, *p_2_*=0.022, BF_10_=6.19). Hence, for Faces, static shape information was sufficient to explain the rating of the videos, while for Hands, the shape information in the frames we selected was not sufficient. It should be noted that, in principle, information contained in other frames may have contained useful information, but informal reports of some participants confirmed that they paid attention more to kinematic than configurational cues.”

Scholarship/Past Work– In several places, particularly in the introduction, a small set of overlapping papers are cited that draw heavily on the authors' prior work, but omit some of the most critical other papers – Krishnan et al. 2016 is not cited at all, and Zhou et al. 2020 is mentioned only in passing, though it is the most similar paper in the literature in design and scope. Corradi dell'Acqua's papers are not cited. Most of the fMRI studies cited in the introduction look at the relatively coarse contrast of observed pain versus control, but do not identify signals that encode the intensity of pain within-person. The studies that have done this most directly are Krishnan et al. 2016 and Zhou et al. 2021. Both identify areas within the mid-insula that encode perceived pain.

We thank the reviewer for this pointer and have edited the introduction to discuss these relevant references more extensively. This section now reads:

“A number of recent studies have used multivoxel pattern analysis to explore how these regions encode the pain of others using fMRI signals, with particular attention to the insula. Krishnan et al. (2016a) showed participants images of hands or feet in painful or innocuous situations and found a pattern across voxels in the insula that could predict how much pain people reported they would feel in the depicted situations. Corradi-Dell’Acqua et al. (2016) reported that the pattern of insula activity could discriminate between trials in which a cue signaled that someone else was receiving a shock from non-shock trials. Finally, Zhou et al. (2020) reanalysed a dataset in which participants viewed photographs of hands in painful or non painful situations or of painful and neutral facial expressions. They found that in the insula similar but dissociable patterns supported painfulness decoding for hands and faces: similar in that a pattern trained to discriminate painfulness from faces could do so from hands and vice versa, with the rostral insula contributing to both patterns; but dissociable because many voxels only contributed to either faces or hands decoding patterns”.

Study Framing– The paper is framed in the introduction around comparisons with direct pain experience, but there is no comparison with pain experience here, so this is somewhat misleading. The most relevant prior papers are not discussed.

We agree that for the study at hand, the overlap with pain experience is not a highly relevant point. We therefore changed the emphasis by including the above-mentioned studies on decoding of perceived pain intensity.

Approach– Key details about the non-insula recordings are omittedA nice feature of the manuscript is the comparison of insular electrodes random electrodes "throughout the brain", but what electrode locations were available, in how many individuals, and what is their distribution? Surely there are not electrode placements everywhere in the brain. Please clarify.

We thank the reviewer for motivating us to indicate the location of all the macro electrode contacts in all patients in the current study. We have now generated a glass brain representation of all macro-contacts available in the 7 patients in Figure 2—figure supplement 2, and a pie chart summarizing the locations covered by the macro contacts in Figure 2—figure supplement 3. The complete list of coordinates can also be found at OSF https://osf.io/mcahz/files/osfstorage/62b97b3702d1f3107cf27c39.

Approach/Results– fMRI study is hard to follow and key details are omittedOne reviewer noted: Section 2.7 is very difficult to follow. I would very much welcome a careful rewrite of this section.Another reviewer wrote that: The concept of the analyses using Neurosynth and fMRI are clever, but the fMRI study in particular is confusing. From the main text it is unclear how many conditions there were in the fMRI study (fMRI during both faces and hands? With ratings?), the sample size, what analyses were done ("leave one subject out" is mentioned but not explained), and whether fMRI activity predicts intensity for either faces or hands. It is stated that "performance did not differ across Hands and Faces", but not that performance was significant for either or, indeed, what analysis was done.

We thank the reviewers for flagging that this section was difficult to follow. We edited the section with these comments in mind and, also mentioned the statistics regarding the Hand or Face conditions in the text, rather than only including them in the figure, as was the case in the previous version. This section now reads:

“Finally, to compare the spatial gradient we find using iEEG with that using fMRI, we leveraged existing data from an unpublished study in our lab using a similar design to measure brain activity using fMRI in healthy participants. Twenty-three independent participants performed an fMRI-adapted version of the rating task. Participants also viewed Hand and Face 2 s videos, presented in different blocks, and were asked to rate them on painfulness. The stimuli depicted the same actor as in the iEEG experiment and were made following the same procedure. To test whether the pattern of BOLD activity in the insula can be used to predict the ratings of the participants, we defined separate regressors in each participants for all trials that participants rates with intensity 0-2, 3-4, 5-6 or 7-8, separately for Hand and Face trials (eight in total). We then performed a partial least-square regression with either two or three components using all voxels in the insula to predict the intensity rating. A leave-one-out cross-validation was used, and the correlation between the predicted and actual rating for each left out participant were compared against zero. This confirmed tht BOLD activity in the insula can be used to predict the perceived intensity for Hands (one sample t-tests for 2 components: *t_(22)_*=3.42, *p_1_*=0.001, BF_+0_=32.32 and 3 components: *t_(22)_*=2.88, *p_1_*=0.004, BF_+0_=10.95) and Faces (one sample t-tests for 2 components: *t_(22)_*=2.78, *p_1_*=0.016, BF_+0_=3.61 and 3 components: *t_(22)_*=3.86, *p_1_*<0.001, BF_+0_=81.35) (Figure 5c), and performance did not differ across Hands and Faces (paired t-test comparing the leave-one-subject out performance for Hands and Faces, with 2 components, *t_(22)_*=0.675, *p_2_*=0.5, BF_10_=0.27; 3 components: *t_(22)_*=-0.39, *p_2_*=0.7, BF_10_=0.23). To compare the spatial pattern of intensity coding across the iEEG and fMRI data, we defined separate regressors for the Hand videos, Face videos, rating-scale and button-presses and used the trial-by-trial ratings given by the participants as parametric modulator, one modulator for the Face and one for the Hand trials, on the respective video regressor. For both Hands and Faces, we found a gradient along the y axis with more anterior locations showing a stronger, and more positive association between BOLD activity and rating (Figure 5d). For Hands, across our 85 bipolar recordings in the patients, locations with higher BBP intensity coding in iEEG also show higher t values in the BOLD signal (Figure 5e). For Faces, on the other hand, we found evidence of absence for an association of the two measures (Figure 5e).”

Approach/Results/Discussion– Unit recordings. Data from 28 unit recordings were provided as additional evidence of pain-of-other coding. This is a very low number of units across only 3 patients, especially given that the results are based on 13 units that showed higher firing during the pain period compared to the baseline period. These observations should probably be relegated to the supplement. At minimum, the wording related to these findings should be revised to reflect the exploratory nature of these results. For example, I do not believe these results warrant labels such as "hand/face specific".

We thank the reviewer for the suggestion. With regard to specificity, we meant the term to reflect that a neuron increases its firing with increasing reported pain intensity for one, but not for the other stimulus set tested, i.e. within our sample of stimuli. To better describe this, we replaced ‘specificity’ throughout the manuscript with ‘preference’, and for the neurons, we now use terms closest to the data, namely Hand-but-not-Face, Face-but-not-Hand or Face-and-Hand to simply reflect the statistics we conducted. Given that *ELife* does not make use of supplementary material sections, moving this data to the supplementary materials is not an option. We hope that in the context of some other papers also reporting small numbers of neurons in humans (e.g. https://www.sciencedirect.com/science/article/pii/S1053811920309848 reporting 14 responsive neurons) readers will still find these neurons a source of information about the response properties of the insula. In the discussion we now specify that the number is small:

“Furthermore, we isolate a small number of insular neurons increasing their firing with increases in the intensity of pain experienced by others”

– Faces vs. hands effects. There do not seem to be significant brain correlations within faces alone that survive correction for multiple comparisons. Figure 3 shows a "trend"-level result, not significant. Could this indicate a "hand vs. face" effect that appears as a correlation in Figure 2? If hands are rated higher than faces, and hands produce greater BBP in the insula, then any electrode that responds to hands more than faces will show up as a correlation between brain and rated intensity. The way to test this would be to test and show correlations within hands and faces separately, but these were apparently not significant for faces. At minimum, the authors need to frankly address this concern in the Discussion

We apologize for confusing the reader with the time-frequency analysis separately for Hand and Face. The main reason we turn to iEEG is to exploit the broadband signal that is closest to neural spiking and inaccessible to conventional EEG. The broadband signal, unlike oscillations, is best analyzed by integrating power over its frequency range. After identifying the relevant BBP range in Figure 2a, which was done including all stimuli not to bias this range towards Hands or Faces (see Kriegeskorte et al., Nature Neuroscience 12 (2009): 535–40), all our statistical inferences in the paper are based on that overall power in that range (20-190Hz) as the proxy of neural activity. Importantly, within that BBP signal, there is significant coding of intensity for Hands and for Faces, after correcting for multiple comparisons in the temporal domain.

The time-frequency decompositions separated for Hand and Face were only meant to illustrate the data that went into the BBP analysis. This time-frequency decomposition is however not ideal to compare conditions, because it inappropriately considers each frequency of the BBP individually, as if these signals were independent, which leads to excessive correction for multiple comparisons and is thus less powered than our preferred BBP analysis that takes the broadband nature of this signal into account. Indeed, in the version of the manuscript we originally submitted, the time-frequency decomposition were in the supplementary materials, not in the main text (see Figure 2 in https://www.biorxiv.org/content/10.1101/2021.06.23.449371v2.full for the original submission). However, the *eLife* editorial team informed us that *eLife* does not support supplementary materials and asked us to include all data into the main manuscript. Accordingly the version seen by the reviewers showed these time-frequency decompositions prominently in Figure 3, and, because it takes much more space to present frequency resolved data than overall BBP data, these illustrations became so large, that they overshadowed the main analysis shown in subpanels 3g,h.

We therefore now moved these time-frequency decompositions into Figure 2—figure supplement 1, without inferential statistics, to focus the manuscript again onto our actual measure of interest, the overall BBP.

Critically, our main analysis (now Figure 2g,h), shows that both for the Hand and for the Face stimuli, there is significant intensity coding, with the relevant inferential statistics. This also addresses the question of whether the correlation observed in Hand and Face could be due to differences between Hand vs Faces, as we see significant intensity coding separately for Hand and for Face stimuli.

To acknowledge the comparatively weaker, although significant, intensity coding amongst our Face stimuli, in the discussion we also write:

“Overall, our ability to decode perceived intensity from our Face stimuli was also lower than that from the Hand stimuli”.

– Faces vs. hands effects. A main point of Figure 3 is that correlations with intensity ratings occur later for faces than hands. But the face signals are not really significant, which makes this kind of conclusion difficult, and there don't seem to be any inferential statistics on the timing of the effect. So it's unclear the timing difference is a real effect. Even though this is "purely explorative" it seems hard to justify an entire main figure and section of the text.

As mentioned in the previous comment, all our inferences are based on the BBP, which do show significant intensity coding for both Hands and Faces. The time-frequency decomposition, that was meant only to be illustrative of the raw data that flows into the BBP calculation, led the attention of the readers astray. We have now moved the time-frequency decompositions to the Figure 2—figure supplement 1 to focus the attention on the BBP analysis that does clearly show significant effects for both stimulus types.

– Faces, hands, and bayes. The facial results appear to hinge on "trend-level" results in several places, and in part on an analysis of lack of differences between faces and hands. It's less desirable to make inferences about the lack of an interaction (a null effect; Figure 3). This is used to argue that there's "no difference between hand and face in the late period". Another interpretation is that there's simply no correlation with intensity for faces. There is some evidence for an effect, but the study perhaps doesn't have enough power to tell for sure. It is helpful that Bayes Factors are reported, but what threshold do the authors think is required to demonstrate evidence for the null in the presence of these multiple tests? (And what is BFincl as opposed to BF10?)

We thank the reviewer for this comment that raises several points we will address separately:

a) As mentioned in the two previous points, our inferences are based on the BBP analyses that do show significant effects separately for hand and face, and we have now moved the distracting time-frequency decompositions – that are indeed underpowered, and also at odds with the very concept of a broadband signal – back into a figure supplement, as it was in the original submission before we were asked to include them into the main manuscript by the editorial team.

b) Power considerations: Because our statistical inferences focus on the overall BBP in two time windows, and we have 60 trials per stimulus, we do have the power to detect modest correlations (a power-analysis reveals that with 60 trials, at α=0.05 and β=0.8, we can detect rho>=0.3 in at least 80% of cases).

c) Evidence for the null: The trial numbers that afford us the power to detect a correlation also afford us the power to provide evidence for the null in a Bayesian framework using the conventional threshold of BF10<⅓, as recommended by Jeffreys 1961. Of course, such a Bayes factor is only relative evidence that the data is 3 times more likely under H0 than H1. In the particular case of the 2 stimuli (Hand vs Face) x 4 ratings Anova, the conclusion that intensity coding was similar for hands and faces is based on BFincl=0.034, showing that the data is 29 times more likely under the hypothesis that the BBP-Rating relation is the same for Hand and Face than under the hypothesis that the relation is different.

d) BFincl: We follow the nomenclature used in JASP and in Rouder et al., 2016, that use BFincl for factorial ANOVA analyses that compare the likelihood of the data in all models including a particular factor (or interaction) against the likelihood of the data in models without that factor (or interaction). This has now been added to the Material and Methods section where we now write:

“When performing Bayesian ANOVAs, we report BFincl which is the likelihood of models including a particular factor (or interaction of factors) divided by the likelihood of models excluding that particular factor (or interaction), as recommended by Rouder and co-workers and implemented in JASP (Rouder et al., 2017, 2016, 2012).”

Discussion– Frank and complete discussion of latencies. The authors report timing effects in the range of 40-320 ms. They cite the well-known study by Krolak-Salmon and colleagues. But it is difficult to know what to make of the current results. On the one hand, latencies on the range of 200-300 ms (across the brain) would be compatible with some studies (but not others). How does one interpret latencies as low as 40-60 ms in the insula, which presumably are not really feasible? On the other hand, the results in the 560-1000 ms range are consistent with many studies but are discussed as "too slow". Authors need to discuss the likely source of these latencies.

We thank the reviewer for encouraging a more extensive discussion of latencies. We tried to identify a wider range of publications that examined latencies in the insula to complex sensory stimuli, and moved the discussion of latencies more towards the Discussion section. Examining the papers we found, we agree that latencies below 100ms are rare (but see Chen et al., 2009 for a 40ms report), with latencies around 150ms more frequent. We also found that although responses sometimes *last* beyond 500ms, we did not find studies that report responses to *start* with latencies above 560ms for visual stimuli – but if the reviewer knows of such examples, that would be valuable to us. Our search therefore continues to reinforce the notion that motion driven responses starting 560ms after the motion signal are not particularly in line with the existing literature. In the Discussion section we therefore now write:

“Leveraging our high temporal and spatial resolution, we found that locations that showed intensity-coding for the Hand stimuli have activity timing echoing the timing of pain-related motion cues, albeit with somewhat unusually relatively short latencies <100 ms, and that the association of motion energy and broadband activity is mediated by the perceived intensity. Such short latencies are below what is typically reported for the onset latency to sensory stimuli in the insula, with most studies showing latencies above 80ms for most sensory stimuli (Bastuji et al., 2018, 2016; Chen et al., 2009; Cornwell et al., 2008; Krolak-Salmon et al., 2003; Liberati et al., 2020, 2016). A particularity of our movie stimuli, however, is that seeing the belt descend upon the hand provides the brain with the kind of temporal context that is known to trigger predictive processes that reduce the response latency of neurons substantially compared to stimuli presented in isolation (Perrett et al., 2009; Reddy et al., 2015). Such predictions could help explain the short latencies we observe here. Locations that show intensity coding for the Face appear to have activity echoing the timing of shape information with latencies in the 40-320 ms range. These latencies are in a range similar to those found in other studies using facial expressions in the insula (Chen et al., 2009; Cornwell et al., 2008; Krolak-Salmon et al., 2003; Meeren et al., 2013). following nociceptive stimulation (Liberati et al., 2020) or static disgusted facial expressions (Chen et al., 2009; Krolak-Salmon et al., 2003). Using automated software to detect the level of activation of the facial action units 4 and 7 (i.e., lowering the eye-brows and tightening the eye-lids), we found that this action unit information suffices to predict participants’ rating of the stimuli with high accuracy, and followed the time course of the neural activity in the Face intensity encoding locations well enough to suggest that it provides traction on the analyses of dynamic pain-related facial expressions. The long neuronal lags we revealed for the facial motion information on the other hand, with the earliest significant associations occurring 560ms after the motion information, are unusually long for the start of responses to sensory stimuli in the insula, as the onset latency for visual, auditory, tactile or nociceptive responses are usually well below 560ms (Bastuji et al., 2018, 2016; Chen et al., 2009; Cornwell et al., 2008; Krolak-Salmon et al., 2003; Liberati et al., 2020, 2016) – although insular activity can persist into such longer intervals (Meeren et al., 2013; Taniguchi et al., 2022). Together this suggests that shape information is more likely than motion information to be the primary driver of the intensity-coding to our facial stimuli”.

We hope that this answers the reviewer’s request.

Discussion/Abstract– Experimental approach does not warrant strong claims of specificity.A Reviewer noted that: A question that one could raise – especially for the hand condition – is whether part of the trial-by-trial correlation between insular activity and ratings was the consequence of a relationship between that activity and low-level motion/shape features of the video clips, rather than a reflection of the insula being involved in "intensity coding for the pain of others". Fortunately, this question was at least partly addresses with the available data, by showing that the recorded insular activity correlated with motion energy when the motion was associated with pain (during the slapping of the belt), but not while the motion was innocuous (during the initial lifting of the belt). Nevertheless, as mentioned by the authors themselves, future experiments using additional control stimuli such as stimuli matched in terms of motion content but differing in the emotions they convey are critical to better understand the specificity of the described insular responses.S/he also noted that: The observation that insular activity correlated with motion energy when motion was associated with pain, and not during the initial lifting of the belt is indeed an indication that the insular responses cannot be reduced to motion detection. However, in these video clips, the first second showed the belt lifting without any movements of the hand. Then, when the belt came down to hit the hand, this was followed by a reaction/movement of the hand. Therefore, one may wonder whether part of the recorded activity could be related to the specific observation of body movement. This might also relate to the observation that stronger insular activity was observed in the hand condition as compared to the face condition.

We fully agree with the reviewer and now write:

“Future experiments using a wider gamut of control stimuli, that are matched in motion but differ in emotions will be critical to address the specificity of the responses we describe, and hence, what state they could reliably signal (Zaki et al., 2016). For the Hand stimuli, this could include seeing hands experience a range of different salient affective experiences, such as a hand being hurt, being caressed and being rejected in addition to a neutral hand-shake (Meffert et al., 2013), or the actor in our Hand movies could have been asked to reposition their hand after each slap.”

A different Reviewer noted that: The study makes use of a fantastic stimulus set from the research group. But if I understand the set correctly, the same actor is used for different levels of pain (face– and hand-based). While this is an excellent set to study some aspects of coding related to stimulus "preference" I don't believe it's general enough to establish "selectivity". Accordingly, I think it would be more reasonable to avoid the claims of "selectivity".

We agree with the reviewer that using only two stimulus types, we can only conclude stimulus-preference for one of the stimulus types over the other, but not stimulus-selectivity in the general sense. We have now replaced all the occurrences of “selectivity” or “specificity” throughout the text to “preference”. The terms “Hand-selective coding” and “Face-selective coding” are also now replaced with “Hand-but-not-Face coding” and “Face-but-not-Hand coding”.

A third Reviewer wrote that: Notably, the authors do not intend to interpret the generalizability to direct pain experience or other affective states, or specificity to pain compared with other affective conditions; but without more information about these, it is difficult to tell what the signals in the insula actually represent. It leaves open the possibility that there is another, better explanation for activation of insular neurons than perceived pain. e.g., It could be about representation of the body more generally, rather than perceived pain specifically. Could they be coding for choices (ratings) themselves? Baliki found that insular activity correlates with rating intensity of a simple visual stimulus, and other studies (e.g., Grinband et al., Neuron) have found that the insula correlates with simple perceptual magnitude decisions. This will be an ongoing project for future work.

We agree with the reviewer and now write:

“An important consideration is whether preference for Face or Hand stimuli in specific locations could simply originate from some participants finding our facial stimuli more salient, and others the hand stimuli, particularly, given that our patients rated the Hand stimuli with slightly higher pain intensity than our Face stimuli. That we find Face and Hand preference side-by-side in simultaneously recorded locations and neurons in single patients suggests that this cannot suffice to explain our data. This is because if a patient were to find Hand stimuli more salient than Face stimuli, and the insula simply codes saliency, we would expect to find Hand-but-not-Face intensity coding in that patient’s insula, but we wouldn’t expect to find side by side locations with Hand-but-not-Face and Face-but-not-Hand coding. This also makes it unlikely that broadband activity simply reflects categorization uncertainty (Grinband et al., 2006). Indeed, it would be uneconomical for all insular neurons to redundantly encode the same exact signal of salience or categorization uncertainty.”

S/he noted that: The stimulus set chosen was limited, and appears to relate to one specific type of painful stimulus on one hand model, and one set of facial expressions made by one female model. As it's well known in general that neurons often have complex receptive fields, it's unclear whether other types of painful hand stimuli or facial expressions made by other models would yield similar findings. Perhaps the insula would respond more strongly to other faces – or perhaps not at all. Other studies have shown that women's pain is discounted (Zhang et al. 2021). The need for diverse sets of stimuli to establish relationships with brain activity that are not stimulus-specific is becoming increasingly recognized (e.g., Yarkoni 2020, Westfall and Yarkoni).

We agree, and now added in the discussion:

“In general, using multiple actors and painful hand interaction could help characterize how intensity coding in the insula is influenced by details of the stimuli that convey it, including the identity and gender of the actor, and how well our results generalize to other stimulus sets (Yarkoni, 2020)”.